# Towards Holistic Multimodal Interaction: An Information-Theoretic Perspective

## Abstract

Multimodal interaction, which assesses whether information originates from individual modalities or their integration, is a critical property of multimodal data. The type of interaction varies across different tasks and subtly influences the effectiveness of multimodal learning, but it remains an underexplored topic. In this paper, we present an information-theoretic analysis to examine how interactions affect multimodal learning. We formulate specific types of information-theoretical interactions and provide theoretical evidence that an effective multimodal model necessity comprehensive learning across all interaction types. Moreover, we analyze two typical multimodal learning paradigms—joint learning and modality ensemble—and demonstrate that they both exhibit generalization gaps when faced with certain types of interactions. This observation underscores the need for a new paradigm that can isolate and enhance each type of interaction. To address this challenge, we propose the *Decomposition-based Multimodal Interaction learning* (DMI) paradigm. Our approach utilizes variation-based decomposition modules to segregate multimodal information into distinct types of disentangled interactions. Then, a new training strategy is developed to holistically enhance learning efficacy across various interaction types. Comprehensive empirical results indicate our DMI paradigm enhances multimodal learning by effectively decomposing and targeted improving the learning of interactions.

## 1 Introduction

A key property of multimodal data is that new information emerges beyond the original unimodality when modalities are presented simultaneously. Accomplishing different tasks requires these modalities to interact in distinct ways. For example, some tasks can be fulfilled by the common part of modalities (*e.g.* vision and audio provide consistent information towards action recognition task Kay et al. (2017)), while certain tasks can only be accurately completed through the integration of multiple modalities (*e.g.*, sarcasm Attardo et al. (2003) derives from the inconsistency between facial expressions and voice). To address this, the concept of interaction is introduced, aiming to describe how multimodal information arises from the integration of modalities. A formalized exploration of these interactions can provide valuable insights into multimodal learning and aid in the development of more effective multimodal models.

Previous studies on multimodal learning have explored interactions from different perspectives. One of the typical studies is to design elaborate fusion architectures that enhance the learning of interactions from the model perspective (Tsai et al., 2019; Nagrani et al., 2021). Other studies focus on specific types of interactions, often presented as relationships among modalities. These approaches regulate and enhance learning among individual modalities, ensuring modality consistency (Cui et al., 2024) and an abundance of overall information (Liang et al., 2024). While these methods empirically consider different interactions, theoretical explanations of multimodal interactions remain unexplored. To solve it, Liang et al. (2023b) introduced an information-theoretic decomposition (Bertschinger et al., 2014) that rigorously quantifies different types of interactions. This framework divides interactions into three categories: Redundancy, Uniqueness, and Synergy. The quantification of these interaction types provides evidence for the choice of multimodal backbone. However, there remains a gap in theoretically explaining how interactions influence multimodal learning.

To fill this gap, we propose information-theoretic analyses to describe and investigate the impact of interaction on multimodal learning. We regard interactions as an inherent property of data. Generally, multimodal data involve a combination of different interactions, which includes three types: **R**edundancy, **U**niqueness, and **S**ynergy. Our analysis begins by lower-bounding the performance of learned multimodal information under different interaction combinations. To further investigate the role of interactions, we examine two typical multimodal training paradigms: joint learning and modality ensemble, assessing their effectiveness across specific interaction combinations. Our results indicate that the joint learning paradigm becomes less effective when dealing with redundant interactions, while the modality ensemble paradigm deteriorates badly when considering synergistic interactions. Besides theoretical analysis, empirical evidence is also provided on synthetic datasets. As shown in Figure 1, we validate methods' performance under different interaction combinations. It can be observed that the performance of modality ensemble decreases significantly with the **S**ynergy interaction (see the left bottom of Figure 1). And joint learning shows worse performance than ensemble mainly under high **R**edundancy (*e.g.* $\frac{1}{4}U + \frac{3}{4}R$). These empirical observations just align with our former theoretical analysis.

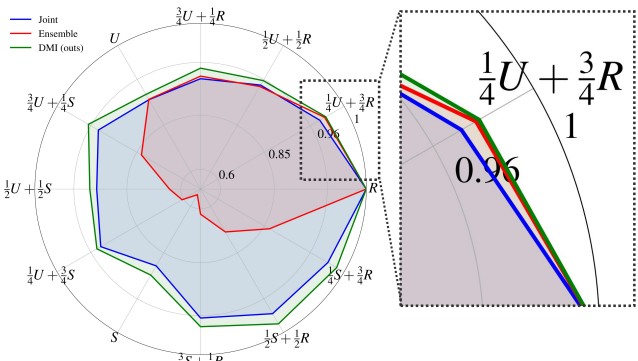

Figure 1: Accuracy of compared paradigm under different predetermined interactions, *e.g.* $\frac{1}{4}U + \frac{3}{4}R$ indicates that $\frac{1}{4}$ of the samples contain **U**nique interactions, while others exhibit **R**edundancy interaction. More analysis is provided in Section 4.3.

Based on the above analysis, we introduce a new learning paradigm, *Decomposition-based Multimodal Interaction learning* (DMI), to holistically improve all types of data interactions. Initially, our method applies variational inference (Alemi et al., 2016) to disentangle multimodal information into types of interactions, including Redundancy, Uniqueness, and Synergy. This decomposition module consists of two sub-modules, Task-related Decomposition and Consistent Decomposition. Then, the disentangled interactions are targetedly learned, to holistically enhance all types of multimodal interaction. Recall Figure 1, our DMI method can consistently achieve superior performance to other multimodal paradigms under different interaction combinations. Besides that, we also conduct extensive experiments to confirm our theoretical analyses and further validate the effectiveness of our new multimodal paradigm. Overall, our contributions are as follows:

1. We propose an information-theoretic analysis focused on data interactions, highlighting the importance of mastering interaction combinations for effective multimodal learning.
2. We examine two prevalent multimodal learning paradigms—joint learning and ensemble learning—and expose their shortcomings in addressing specific types of interactions.
3. We introduce a new paradigm that improves multimodal learning, by decomposing and holistically enhancing multimodal interactions.

In this paper, we revisit multimodal interactions from the information-theoretic perspective. Beyond pure quantification in existing studies, each type of interaction is disentangled and targetedly enhanced in our new. This new multimodal learning paradigm shows its efficacy with holistic multimodal interactions.

## 2    RELATED WORK

### 2.1    PROGRESS IN MULTI-MODAL LEARNING

With the advancement of various modalities, multimodal learning has garnered increasing attention. Its applications now extend to areas such as action recognition (Soomro et al., 2012), video understanding (Chen et al., 2020), and sentiment analysis (Zadeh et al., 2016). To tackle these tasks and

effectively leverage the rich information contained in different modalities, previous methods have developed diverse architectures for extracting and fusing multimodal data, demonstrating impressive performance (Nagrani et al., 2021; Kim et al., 2021). Despite these advances in modeling, deeper explanations and analyses of multimodal learning are still needed. Theoretical frameworks have been proposed to clarify why multimodal models outperform unimodal ones, focusing on aspects such as representation (Huang et al., 2021) and learning difficulty (Lu, 2023). Additionally, research on the modality imbalance problem (Wang et al., 2020; Zhang et al., 2024), where multimodal models tend to over-rely on certain dominant modalities Wang et al. (2020), has provided insights into more effective multimodal learning strategies Peng et al. (2022); Fan et al. (2023). Most of these findings and innovations are model-centric, however, the differences in the information contained within the data itself also lead to variations in learning. In this paper, we investigate how information emerges when multiple modalities intrinsically interact from a data-centric perspective and describe its impact on multimodal learning.

## 2.2 INFORMATION THEORY FOR MULTIMODAL LEARNING

Information theory provides a powerful framework for understanding the model learning process, particularly when dealing with multiple data sources. A key focus is how information is extracted and combined from each source. On one hand, some researchers enhance the consistency (Federici et al., 2020; Cui et al., 2024) guided by information bottleneck, while others improve the richness of joint information by minimizing redundancy across modalities (Liang et al., 2024). On the other hand, certain methods seek to quantify how well a model captures multimodal interactions (Liang et al., 2023b), drawing from information decomposition approaches (Bertschinger et al., 2014). This model-centric interaction can be further expanded and measured, offering valuable intuition for model selection and evaluation (Liang et al., 2023a). In this paper, we take one step further, explaining and addressing the impact of various interactions on multimodal learning.

## 3 METHOD

### 3.1 PRELIMINARY

We consider the universally-used two modalities situation $(X^{(1)}, X^{(2)})$ as multimodal data and $Y$ as the ground truth. The multimodal learning paradigm processes each unimodality $X^{(1)}, X^{(2)}$ into features $Z^{(1)}, Z^{(2)}$ using unimodal encoders $\phi^{(1)}$ and $\phi^{(2)}$, respectively, and combines them in some way to obtain the predicted output $\hat{Y}$, as shown in Figure 2 (a). Generally speaking, the multimodal information after fusion is greater than the maximum of the original information (Huang et al., 2021). This is intuitive since the multimodal feature fuse information from all modalities. This concept can be expressed as:

$$I(Z^{(1)}, Z^{(2)}; Y) = I(Z; Y) \geq \max\left(I(Z^{(1)}; Y), I(Z^{(2)}; Y)\right), \tag{1}$$

where $Z = \{Z^{(1)}, Z^{(2)}\}$, and $I(Z; Y) = \int p(z, y) \log \frac{p(z,y)}{p(z)p(y)} \, dz \, dy$ represents the mutual information between the variables $Z$ and $Y$.

### 3.2 INFORMATICS PERSPECTIVE OF MULTIMODAL LEARNING

Machine learning aims to explore the relation between inputs and the target. Typically, this relation is often quantified by mutual information and higher mutual information implies a lower error rate (Morishita et al., 2022). For example, learning for $m$-th modality aims to enlarge the mutual information between representation $Z^{(m)}$ and target $Y$. According to the information processing inequality (Beaudry & Renner, 2011), the unimodal information is subject to the following constraint:

$$I(X^{(m)}; Y) \geq I(Z^{(m)}; Y), m \in [2]. \tag{2}$$

This indicates that the learned information is always less than the data-side information (Xu et al., 2020). Extending to multiple modalities, a similar inequality holds:

$$I(X^{(1)}, X^{(2)}; Y) = I(X; Y) \geq I(Z; Y) = I(Z^{(1)}, Z^{(2)}; Y). \tag{3}$$

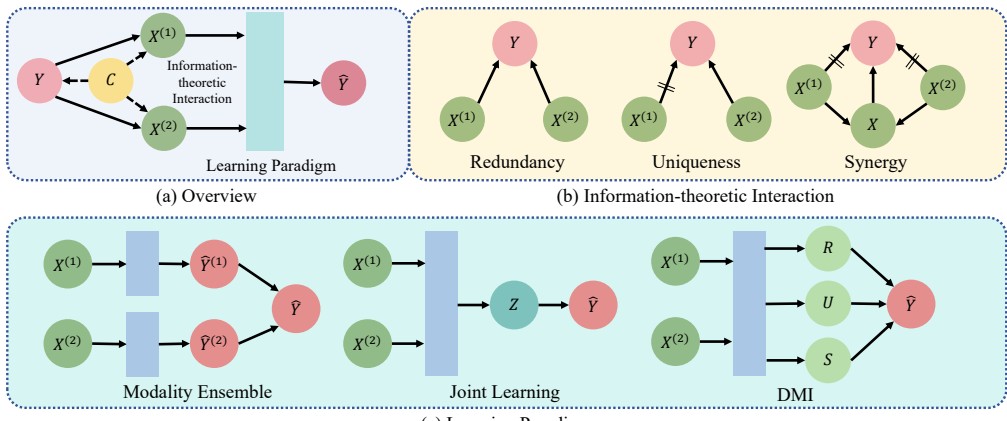

Figure 2: (a) An overview of multimodal learning framework. (b) Information-theoretic Interaction describes the relation from data with two modalities $X^{(1)}, X^{(2)}$ to target $Y$, consists of Redundancy, Uniqueness, and Synergy. (c) The Learning Paradigm compares typical multimodal learning paradigms, Joint Learning and Modality Ensemble, with the DMI Learning paradigm, which decomposes the interaction and uses it to guide learning accordingly and holistically.

Equation 3 presents the upper bound for the information learned by the model. From this perspective, multimodal learning aims to maximize the mutual information between the representation $Z$ and the target $Y$, thereby approaching the information within multimodal data as closely as possible. However, there is still a lack of acknowledge of how multimodal information is derived and influences multimodal learning.

To address it, we propose the concept of interaction to address the critical question: *How is multimodal information developed from the integration of modalities?* To achieve this, we divide the multimodal information into three types of interactions: **Redundancy**, **Uniqueness**, and **Synergy** (see Figure 2 (b)). Redundancy occurs when the shared components among modalities can accomplish the task. Uniqueness arises when only one modality can relate to the target while the other cannot. Synergy occurs when both modalities, though insufficient on their own, jointly emerge new information for completing the task. And the multimodal information is the combination of information from these interactions. Accordingly, we define $\boldsymbol{c} \in C \subseteq \mathbb{R}^4$ as the *interaction combination*, indicating the proportion of each interaction in the multimodal information. $c_1$ corresponds to redundancy, $c_2$ represents uniqueness for modality (1), $c_3$ represents uniqueness for modality (2), and $c_4$ denotes synergy. The combination is subject to the constraints $\sum_i c_i = 1$ and $c_i \geq 0$ for $\boldsymbol{c} \in C$. We hope to consider the data of different interaction combinations separately to facilitate analysis. Hence, $\boldsymbol{c}$ disassembles samples into different categories based on their interaction combinations:

$$p(x, y, \boldsymbol{c}) = \begin{cases} p(x, y) & \text{Inter}(x, y) = \boldsymbol{c}, \\ 0 & \text{otherwise.} \end{cases} \tag{4}$$

$\text{Inter}(x, y) = \boldsymbol{c}$ constraint that the proportion of Redundancy, Uniqueness, and Synergy information inside $(x, y)$ accords with $\boldsymbol{c}$. With the above definition, we can derive the following bound, which presents the importance of learning about different interaction combinations in multimodal learning.

**Proposition 3.1.** *Let $\boldsymbol{c}$ be the interaction combination, and let $I(Z; Y)$ be modeled by a multimodal model. The following inequality holds:*

$$I(Z; Y) \geq E_{\boldsymbol{c}}[I(Z; Y|\boldsymbol{c})] + \zeta, \tag{5}$$

*where $\zeta$ is a constant that is independent of the model.*

The proof is provided in Section A.1. Proposition 3.1 provides a lower bound using an expectation of learned information across various interaction combinations, conditioned on $\boldsymbol{c}$. Hence, for a fixed $\boldsymbol{c}$, increasing the information under this combination $\boldsymbol{c}$ can enhance the lower bound. Therefore, When the model exhibits holistic learning across information of diverse interactions, the learned multimodal information $I(Z; Y)$ can be ensured. Accordingly, ensured multimodal information can lead to a lower error rate for multimodal learning (Morishita et al., 2022).

| $|I - I_S|$ | Redundancy | Uniqueness | Synergy |
|---|---|---|---|
| Joint | $\leq \xi + \sqrt{\omega/n}$ | $\leq \xi + \sqrt{\omega/n}$ | $\leq \xi + \sqrt{\omega/n}$ |
| Ensemble | $\leq \xi + \sqrt{\omega/(2n)}$ | $\leq \xi + \sqrt{\omega/n}$ | $\geq \max\left(I_S^{syn}(Z^{(1)}; Y), I_S^{syn}(Z^{(2)}; Y)\right)$ |

Table 1: Learning objectives and generalization gaps under different interaction for joint learning and modality ensemble paradigms.

### 3.3 IMPACT ON LEARNING UNDER INTERACTIONS

To analyze the impact of interactions, we investigate two typical multimodal learning paradigms: joint learning and modality ensemble (see Figure 2 (c)). In detail, joint learning integrates all modalities to jointly learn the prediction, while modality ensemble only explores the information within each modality and integrates unimodal predictions. Both paradigms share the same hypothesis space to facilitate comparison and differ in training objectives. Considering the widely used cross-entropy loss, the objective of *joint learning* with training set $S$ with $n$ samples can be denoted as:

$$\min \frac{1}{n} \sum_{i=1}^{n} [-\log p(y_i | z_i^{(1)}, z_i^{(2)})] = \max I_S(Z; Y) - H_S(Y). \tag{6}$$

And the objective of *modality ensemble* with the same training set $S$ can be denoted as:

$$\min \frac{1}{n} \sum_{i=1}^{n} [-\log p(y_i | z_i^{(m)})] = \max I_S(Z^{(m)}; Y) - H_S(Y), \ m \in [2], \tag{7}$$

where $I_S(Z^{(m)}; Y) = \frac{1}{n} \sum_{i=1}^{n} [\log(p(y_i | z^{(m)})/p(y_i)]$ denote the average empirical mutual information over the dataset, and $H_S(Y) = \frac{1}{n} \sum_{i=1}^{n} [-\log p(y_i)]$ stays constant.

To illustrate how well different paradigms learn, we focus on the generalization gap in information (Xu et al., 2020), which describes the difference in information between training data and unseen data drawn from the same distribution. We introduce the following proposition.

**Proposition 3.2.** *Let $g$ be an estimator that maps samples $z$ and target $y$ to their pointwise mutual information. For a training set $S$ with interaction $c$, the estimator learned is denoted as $g_S$. Assume that for all $g \in \mathcal{G}, z \in \mathcal{Z}, y \in \mathcal{Y}$, the value $g(z, y)$ is bounded within $[-B, B]$. For any $\delta > 0$, with at least a probability of $1 - \delta$, the following inequality holds for each $g_S$:*

$$|I(Z; Y | \boldsymbol{c}) - I_S(Z; Y | \boldsymbol{c})| \leq 2\Re_S(\mathcal{G}) + B\sqrt{\frac{\log(1/\delta)}{2N}}, \tag{8}$$

*where $N$ denotes the number of training samples, and $\Re_S(\mathcal{G})$ denotes the empirical Rademacher complexity, which measures the richness of the hypothesis class.*

This theorem (Equation 8) illustrates the generalization gap between the mutual information estimated from the training dataset and that from the overall distribution. A narrower gap can ensure that the mutual information inferred during training adequately represents the overall distribution. As we assume that both training paradigms share the same hypothesis space, they have the same Rademacher complexity. Hence, we can denote $\xi = 2\Re_S(\mathcal{G})$ and $\omega = \sqrt{B^2 \log(1/\delta)/2}$. This notation can emphasize the importance of the scale of training samples to reduce this gap.

In the following section, we will provide an analysis of different multimodal interactions in the above two typical paradigms, joint learning and modality ensemble. In this section, we use superscripts to denote the types of different latent interactions. For example, $I^{red}(\cdot) = I(\cdot | \boldsymbol{c} = [1, 0, 0, 0])$ denotes the interaction is completely redundancy.

**Redundancy.** Both modalities of samples with Redundancy interactions demonstrate consistency with the target. The information on redundant interaction can be defined as:

$$I^{red}(X^{(1)}, X^{(2)}; Y) = I^{red}(X^{(1)}; Y) = I^{red}(X^{(2)}; Y) = H(Y). \tag{9}$$

The joint learning method complies with a hypothesis in Proposition 3.2, for $n$ distinct samples are utilized for training. For joint learning, the upper bound in Proposition 3.2 becomes $\xi + \sqrt{\omega/n}$.

Although modality ensemble also obeys this hypothesis, its objective for each unimodality help learn more information under redundant interactions. That inspires us to determine a tighter upper bound to better describe the generalization gap. We propose the following lemma:

**Lemma 3.3.** *The main difference between multimodal joint training and modality ensemble lies in the number of samples, with the ensemble providing a tighter upper bound with $\xi + \sqrt{\omega/(2n)}$.*

The analysis is provided in Section A.3. Hence, by leveraging the redundant information across modalities, the modality ensemble can exploit this surplus to achieve better learning outcomes. In contrast, joint learning, which models the integration of modalities, falls short in fully exploiting the information within each modality, resulting in a slightly larger generalization gap, as illustrated in the first column of Table 1.

**Uniqueness.** Interactions of uniqueness broadly occur where only one specific modality is capable of completing the task. Without loss of generality, the information concerning unique interactions in modality 1 can be defined as follows:

$$I^{uni1}(X^{(1)}, X^{(2)}; Y) = I^{uni1}(X^{(1)}; Y) = H(Y), \ I^{uni1}(X^{(2)}; Y) = 0. \tag{10}$$

For joint learning, the approach involves learning from $n$ samples through multimodal integration, accord with the setting of Proposition 3.2, thus have an upper bound with $\xi + \sqrt{\omega/n}$. The modality ensemble learns from $n$ samples, and it also accords with the setting of Proposition 3.2. Therefore, the upper bound on the generalization gap is $\xi + \sqrt{\omega/n}$. Both paradigms exhibit comparable performance in terms of unique interactions, as shown in the second column of Table 1.

**Synergy.** Each unimodal data with synergy interactions inherently lacks information pertinent to the target by itself. However, the integration of these modalities results in the emergence of additional information for the target. Data with synergy interactions can be defined as:

$$I^{syn}(X^{(1)}, X^{(2)}; Y) = H(Y), \quad I^{syn}(X^{(1)}; Y) = I^{syn}(X^{(2)}; Y) = 0. \tag{11}$$

For joint learning, since it can learn from the integration of modalities, $n$ data are sufficient for this approach to learning the information, which also aligns with the setting of Proposition 3.2. Thus, joint learning has a upper bound of $\xi + \sqrt{\omega/n}$. For modality ensemble, this approach—modeling each modality independently—fails to effectively find the relation between each unimodality and the target at hand. Therefore, we re-examine the generalization gap for modality ensemble under conditions of synergy interactions:

$$|I^{syn}(Z; Y) - I_S^{syn}(Z; Y)| \geq \max\left(I_S^{syn}(Z^{(1)}; Y), I_S^{syn}(Z^{(2)}; Y)\right). \tag{12}$$

The detailed analysis is provided in Section A.4. This inequality establishes a lower bound for the generalization gap, highlighting the inadequacy of the modality ensemble under conditions of synergy interactions. Consequently, modality ensemble performs poorly in scenarios involving synergy interactions, which accords with the third column of Table 1.

Overall, our analysis identifies two discrepancies in interaction learning within the typical multimodal paradigm. Specifically, joint learning becomes less effective when dealing with redundant interactions compared to modality ensemble. Conversely, modality ensemble experiences a significant performance loss under synergy types of interactions, as it lacks the joint information among modalities. According to Proposition 3.1, the failure of the previous paradigm to effectively handle certain interaction combinations results in a decrease in the lower bound, adversely affecting the overall error rate in multimodal learning. Consequently, there is an urgent need for a paradigm that facilitates holistic interaction learning.

### 3.4 MODULATIONS FOR VARIOUS INTERACTION

In this section, we propose a new paradigm for multimodal learning, named *Decomposition-based Multimodal Interaction learning* (DMI), to achieve holistic learning for each type of interactions. In detail, we propose a decomposition-based module to explicitly determine different interactions with Redundancy, Uniqueness, and Synergy (see Figure 3 (a)), and further design a three-step training strategy to help achieve decomposition and improved learning.

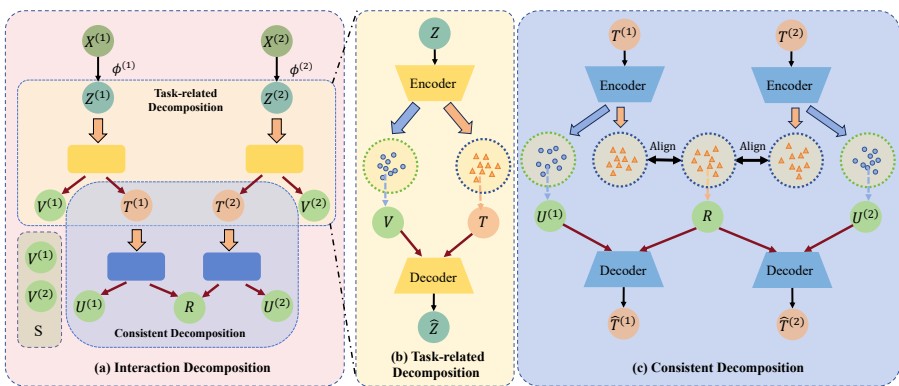

Figure 3: (a) Overall illustration of our proposed *Interaction Decomposition Module*. (b) Task-related decomposition applies a variation-based decomposition to extract the task-related information for each modality. (c) Consistent Decomposition applies to task-related variables and separates consistent interactions $R$ from specific ones $U^{(1)}, U^{(2)}$.

First, we propose a decomposition method to distinguish different types of interactions within the data. We begin by determining whether each unimodal representation contains sufficient information to complete the task. As defined in Section 3.3, data with redundancy or uniqueness interactions can provide relevant information for the task, whereas data with only synergy interactions do not contribute directly to the task. To address this, we introduce a task-related decomposition module, which decomposes each unimodal representation $Z^{(m)}$ into two components: task-related $T^{(m)}$ and task-irrelevant $V^{(m)}$ latent variables (See Figure 3 (b)). To ensure that these variables are both informative and disentangled, we apply an objective function inspired by the Variational Information Bottleneck (Alemi et al., 2016), which is formulated as:

$$\max -I(T^{(m)}, V^{(m)}) = I(Z^{(m)}; T^{(m)}, V^{(m)}) - I(Z^{(m)}; T^{(m)}) - I(Z^{(m)}; V^{(m)}), \ m \in [2]. \quad (13)$$

We discuss the decomposition objective in Equation 13 in more detail in Section A.5. This objective controls the flow of information from the unimodal representation $Z^{(m)}$ into two distinct components: $T^{(m)}$ (task-related) and $V^{(m)}$ (task-irrelevant). When this objective is properly optimized, $T^{(m)}$ and $V^{(m)}$ will encapsulate distinct parts of the information, thereby achieving disentanglement. Since $V^{(1)}$ and $V^{(2)}$ individually contribute little information to the task, the information emerging from the integration of $V^{(1)}$ and $V^{(2)}$ can be interpreted as a synergy interaction, similar to the concept outlined in Equation 11.

After obtaining the task-related features $T^{(1)}$ and $T^{(2)}$, we further decompose them into unique and redundant interactions. The difference is that redundant interaction shows consistency among modalities, while uniqueness shows the specificity of each modality. Thus, this distinguishing can be achieved by decomposing the consistency and specific parts (Hwang et al., 2020). Specifically, we decompose the task-related information $T^{(1)}, T^{(2)}$ into three components: $U^{(1)}, U^{(2)}$ and $R$, where $U^{(m)}$ represents uniqueness, while $R$ represents redundancy (See Figure 3 (b)). Thus, the objective of the consistent decomposition can be denoted as:

$$\max \ 2I(T^{(1)}; T^{(2)}; R) - I(U^{(1)}; R) - I(U^{(2)}; R), \quad (14)$$

where the mutual information $I(U^{(m)}; R)$ can be reformulated using the idea of Equation 13. Then, these interactions can be decomposed by these module designs. The detailed architecture is described in Appendix B.2. Furthermore, a new training strategy is proposed to improve decomposition quality and learn targeted interactions. This strategy contains three-stage:

1. In the early stages of learning, the unimodal representation $Z^{(m)}$ acquires only limited information, which makes decomposition challenging. To address this, we *warm up* each unimodal encoder by applying the ensemble objective Equation 32 for several epochs, ensuring that each unimodal encoder obtains the necessary information for decomposition.

2. After warm up, we *freeze* each **unimodal encoder** $\phi^{(m)}$ and focus solely on training the decomposition module, to stabilize the training process.

| | Dataset | CREMA-D | | Kinetic-Sound | | UCF101 | |
|---|---|---|---|---|---|---|---|
| | Metric | ACC | F1 | ACC | F1 | ACC | F1 |
| Unimodal | Visual/RGB | 41.4 | 41.2 | 74.3 | 74.2 | 76.9 | 76.1 |
| | Audio/OF | 65.3 | 65.7 | 66.2 | 65.8 | 67.8 | 67.6 |
| Baseline | Joint | 70.2 | 71.0 | 84.1 | 84.1 | 78.8 | 78.0 |
| | Ensemble (Du et al., 2023) | 68.8 | 69.5 | 86.0 | 85.9 | 82.3 | 81.8 |
| Regulation | OGM (Peng et al., 2022) | 70.2 | 71.0 | 84.1 | 84.1 | 78.9 | 78.0 |
| | PMR (Fan et al., 2023) | 71.1 | 71.5 | 85.8 | 85.3 | 77.6 | 76.6 |
| | AGM (Kontras et al., 2024) | 70.2 | 71.0 | 84.4 | 84.1 | 80.2 | 79.8 |
| Interactions | MMTM (Vaezi Joze et al., 2020) | 70.4 | 71.0 | 85.4 | 85.3 | 78.6 | 78.3 |
| | MIB (Mai et al., 2022) | 71.1 | 71.7 | 84.7 | 84.6 | 83.8 | 83.3 |
| | CEN (Wang et al., 2022) | 69.1 | 69.6 | 84.5 | 84.3 | - | - |
| | MMdyn (Han et al., 2022) | 69.8 | 70.2 | 65.3 | 65.6 | 77.1 | 76.5 |
| | QMF (Zhang et al., 2023) | 68.4 | 69.1 | 84.3 | 84.2 | 77.2 | 76.5 |
| | MMML (Wu et al., 2024) | 70.8 | 71.8 | 83.2 | 83.1 | 79.3 | 78.9 |
| Ours | DMI | **73.1** | **73.8** | **86.8** | **86.7** | **84.2** | **83.9** |

Table 2: Validation on various **CNN-based** multimodal interaction method on audio-visual datasets, CREMA-D, Kinetic-Sounds, and UCF101. Best results are presented in bold.

3. After interactions have been decomposed, we jointly integrate them into the target space to finetune the entire model. This process enhances the holistic learning of these interactions, enabling the model to better capture and utilize the information across modalities.

Overall, our DMI paradigm can explicitly decompose different interactions, holistically enhance the learning of each interaction, and further achieve more effective multimodal learning.

## 4 EXPERIMENT

### 4.1 EXPERIMENT SETTING

**Datasets  CREMA-D** (Cao et al., 2014): An emotion recognition dataset with two modalities (audio and visual), covering six emotions: anger, happiness, sadness, neutrality, disgust, and fear. It contains 7,442 video clips. **Kinetic-Sounds** (Arandjelovic & Zisserman, 2017): A multimodal action recognition dataset with audio and visual modalities. It includes 19,000 ten-second clips from 31 human action classes selected from the Kinetics dataset. **CMU-MOSEI** (Zadeh et al., 2018): A multimodal dataset for sentiment analysis and emotion recognition, incorporating audio, visual, and text modalities. It contains 23,453 annotated video segments sourced from YouTube. **UCF101** Soomro et al. (2012): A multimodal action recognition dataset with RGB and optical flow modalities. It includes 13,320 videos from 101 human action classes, often used for video synthesis and prediction tasks.

**Backbone**: In the CNN-based experiments, we use ResNet18 (He et al., 2016) as the backbone, modifying the first layer's channel number to accommodate each modality. Specifically, for the audio modality, we set the channel to 1, and for optical flow, the channel is set to 2. In the Transformer-based experiments, for Kinetic-Sound dataset, images and audio are encoded using a Vision Transformer (ViT). For CMU-MOSEI dataset, we follow the preprocessing steps outlined in Liang et al. (2021) and apply Transformer to encode different modalities (Liang et al., 2021).

### 4.2 COMPARISON ON REAL-WORLD DATASETS

To demonstrate the advantages of our *Decomposition-based Multimodal Interaction learning* (DMI) paradigm, we conducted comparisons with previous multimodal methods on real-world datasets. The comparison methods can be divided into three categories: (1) *Baselines*, which include typical multimodal learning methods such as joint learning and unimodal ensemble learning; (2) methods

| | Dataset | Kinetic-Sound | | CMU-MOSEI | | | |
| | Modality | Audio+Visual | | Audio+Text | | Visual+Text | |
| | Metric | ACC | F1 | ACC | F1 | ACC | F1 |
|---|---|---|---|---|---|---|---|
| Unimodal | Audio | 50.5 | 50.3 | 44.2 | 32.1 | - | - |
| | Visual | 50.9 | 50.5 | - | - | 47.3 | 40.7 |
| | Text | - | - | 59.9 | 60.0 | 60.4 | 60.7 |
| Baseline | Joint | 67.9 | 67.6 | 61.9 | 62.2 | 63.1 | 63.0 |
| | Ensemble (Du et al., 2023) | 69.3 | 69.2 | 61.8 | 61.8 | 63.2 | 63.1 |
| Regulation | OGM (Peng et al., 2022) | 68.9 | 68.6 | 61.9 | 62.2 | 63.1 | 63.0 |
| | PMR (Fan et al., 2023) | 68.0 | 68.2 | 62.1 | 62.2 | 62.7 | 62.5 |
| | AGM (Kontras et al., 2024) | 68.9 | 69.1 | 61.9 | 62.2 | 63.1 | 63.0 |
| Interactions | MBT (Nagrani et al., 2021) | 69.9 | 69.9 | 62.0 | 62.2 | 63.0 | 63.2 |
| | MIB (Mai et al., 2022) | 63.1 | 62.9 | 61.9 | 62.1 | 62.2 | 62.0 |
| | QMF (Zhang et al., 2023) | 70.6 | 70.3 | 61.5 | 61.7 | 62.9 | 62.9 |
| | MMML (Wu et al., 2024) | 65.3 | 65.5 | 61.7 | 61.8 | 62.8 | 62.5 |
| Ours | DMI | **70.8** | **71.4** | **63.1** | **63.2** | **63.4** | **63.4** |

Table 3: Validation on **Transformer-based** multi-modal interaction methods on Kinetic-Sounds (Audio+Visual), CMU-MOSEI (Audio+Text), (Visual+Text). Best results are presented in bold.

enhancing multimodal learning through *regulation* on the learning of unimodality (including OGM (Peng et al., 2022), PMR (Fan et al., 2023), and AGM (Kontras et al., 2024)); and (3) specifically designed architectures that intuitively capture *interactions* (including MMTM (Vaezi Joze et al., 2020), MBT (Nagrani et al., 2021), CEN (Wang et al., 2022), MIB (Mai et al., 2022), MMdynamic (Han et al., 2022), MMML (Wu et al., 2024) and QMF (Zhang et al., 2023)). The details of these methods are listed in the Appendix. The backbone is based on CNN Table 2 and Transformer Table 3, respectively. '-' represents that the experiment could not be extended to this dataset.

Based on these empirical results, we have made the following observations. Firstly, joint learning performs better than ensemble learning only in certain settings, which depend on the data and architecture. Secondly, both the unimodal regulation methods and architecture-based interaction learning methods often enhance multimodal learning more than joint learning; however, these improvements are not always consistent across different settings. Under the Kinetic-Sounds dataset, the ensemble method outperforms most of the other comparison methods, possibly because the data is more prone to exhibit interactions that are easier to learn. Thirdly, compared to these methods, our DMI network consistently outperforms all other methods under both backbones, attributed to the fact that our approach can effectively decompose and holistically learn from different interactions. Lastly, our method maintains improved performance when switching the backbone to the Transformer architecture and on text-related tasks, demonstrating the universality of our approach.

### 4.3 INTERACTION VALIDATION ON SYNTHETIC DATASET

**Interaction Learning Performance.** Our analysis and method design focus on learning from different types of interactions, which are often difficult to distinguish and measure directly from the original data. To address this, we construct synthetic datasets in which the intrinsic interactions within the data can be manually configured. In this setup, each sample contains a specific type of data—Redundancy (R), Uniqueness (U), or Synergy (S)—based on how they relate to the target. For redundancy data, both modalities map to the target, whereas for uniqueness data, only one modality is predictive. In synergy data, each modality contributes partial information (similar to the XOR condition (Bertschinger et al., 2014)). We select different datasets, each containing single interactions or two mixtures. We then validated our method against two baseline approaches: Joint learning and Ensemble Learning. As shown in Figure 1, the results for synergy and redundancy data align with our analysis; joint training shows significant improvement in synergy-related data, while unimodal ensemble performs better with redundancy types of interactions. Our method enhances the learning of interactions, significantly outperforming both baseline methods across holistic interactions.

| Task | AND+XOR | | | | OR+XOR | | | | AND+OR+XOR | | | |
|---|---|---|---|---|---|---|---|---|---|---|---|---|
| Measure | $R$ | $U_1$ | $U_2$ | $S$ | $R$ | $U_1$ | $U_2$ | $S$ | $R$ | $U_1$ | $U_2$ | $S$ |
| CVX | 21.4 | 0.0 | 0.3 | 80.9 | 21.2 | 0.0 | 0.6 | 78.3 | 33.7 | 0.4 | 0.1 | 65.8 |
| DMI | 18.9 | 0.2 | 0.0 | 78.2 | 20.8 | 0.0 | 2.4 | 76.9 | 27.7 | 4.8 | 0.0 | 67.6 |
| Truth | 19.1 | 0.0 | 0.0 | 80.9 | 19.1 | 0.0 | 0.0 | 80.9 | 25.5 | 0.0 | 0.0 | 74.5 |

Table 4: Validation of learning interaction from complex Boolean logical relation.

**Interaction Learning demonstration.** Although Section 4.2 demonstrates the effectiveness of our DMI model, it is essential to further validate whether the interactions are well learned by our proposed training strategy. Initially, we constructed datasets where the data were derived from bit-wise features that follow logical relationships with the labels. These include mixtures such as 1/2 AND and 1/2 XOR, 1/2 OR and 1/2 XOR, and 1/3 AND, 1/3 OR, and 1/3 XOR. For comparison, we use the interaction estimator, CVX estimator (Liang et al., 2023b), to estimate the RUS (Redundancy, Uniqueness, and Synergy), whose sum is rescaled to 1. For our DMI method, we estimate the interaction by evaluating the loss of features corresponding to redundancy, uniqueness, and synergy. The experimental results are shown in Table 4. Although our method is not specifically designed for interaction quantification, it effectively learns the interactions in a spontaneous manner. Our framework achieves results comparable to the CVX method, which is specifically designed for quantifying interactions, thereby validating the effectiveness of the proposed decomposition framework. In datasets containing both AND and XOR interactions, our method provides a closer approximation to the ground truth, indicating that our approach remains effective in capturing interactions in complex scenarios.

## 4.4 ABLATION STUDY

In this section, we conduct an ablation study to evaluate the necessity and efficacy of each part of the interaction decomposition module within our framework. The study addresses two primary questions: *1) Is the use of variational methods indispensable? 2) Are both decomposition modules essential for the framework's performance?* In the experiments, we introduce *DMI-Fully Connected* (DMI-FC), which realizes decomposition using fully connected layers instead of variational information bottlenecks. Additionally, we explore configurations that omit either the *Task-related Decomposition* (TD) module or the *Consistent Decomposition* (CD) module. Specifically, DMI-TD retains only the TD module, while DMI-CD preserves the CD module, depicted in Figure 4.

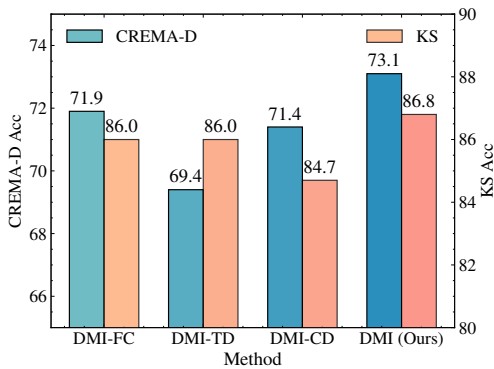

Figure 4: Ablation studies of DMI paradigm on CREMA-D and Kinetic-Sounds (KS) datasets.

On the one hand, we observe that the performance of DMI-FC is higher than that of the other two ablation settings. This is because DMI-FC incorporates the full decomposition process, whereas DMI-TD and DMI-CD only retain partial decomposition modules. On the other hand, DMI significantly outperforms DMI-FC. This improvement is due to the fact that the decomposition in DMI is based on variation, allowing it to decouple different interactions effectively. As a result, DMI can adaptively learn more accurate interactions, leading to better performance. These findings highlight the critical importance of variational interaction decomposition modules within our framework.

## 5 CONCLUSION

We introduce an information-theoretic framework that highlights the importance of learning from different interaction combinations. Additionally, we analyze how typical multimodal learning paradigms—joint learning and modality ensemble—are influenced by specific interaction types. Based on this analysis, we propose a *Decomposition-based Multimodal Interaction learning* (DMI) paradigm that effectively distinguishes and interprets interactions within the data, and we design a three-stage learning process to achieve improved performance.

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

## A  PROOF AND ANALYSIS

### A.1  PROOF FOR PROPOSITION 3.1

**Proposition A.1.** *Let $c$ be the interaction combination, and let $I(Z;Y)$ be modeled by a multimodal model. The following inequality holds:*

$$I(Z;Y) \geq E_{\boldsymbol{c}}[I(Z;Y|\boldsymbol{c})] + \zeta, \tag{15}$$

*where $\zeta$ is a constant that is independent of the model.*

*Proof.* Using the definition of mutual information and the conditional mutual information, we have:

$$I(Z;Y) = \int p(z,y) \log \frac{p(z,y)}{p(z)p(y)} \, dz dy = \int p(z,y) \log \frac{\mathbb{E}_{\boldsymbol{c}} p(z,y|\boldsymbol{c})}{p(z)p(y)} \, dz \, dy$$

$$I(Z;Y|\boldsymbol{c}) = \int p(z,y|\boldsymbol{c}) \log \frac{p(z,y|\boldsymbol{c})}{p(z|\boldsymbol{c})p(y|\boldsymbol{c})} \, dz \, dy. \tag{16}$$

We can obtain the difference as:

$$I(Z;Y) - \mathbb{E}_{\boldsymbol{c}} I(Z;Y|\boldsymbol{c}) = \int p(z,y,\boldsymbol{c}) \left( \frac{\log \mathbb{E}_{\boldsymbol{c}} p(z,y|\boldsymbol{c})}{\log p(z,y|\boldsymbol{c})} + \log \frac{p(z|\boldsymbol{c})p(y|\boldsymbol{c})}{p(z)p(y)} \right) \, dz \, dy \, d\boldsymbol{c} \tag{17}$$

For the former term presented in Equation 17, we can utilize the Jensen inequality:

$$\int p(z,y,\boldsymbol{c}) \frac{\log \mathbb{E}_{\boldsymbol{c}} p(z,y|\boldsymbol{c})}{\log p(z,y|\boldsymbol{c})} dz \, dy \, d\boldsymbol{c} = \int p(z,y|\boldsymbol{c}) \left( \mathbb{E}_{\boldsymbol{c}} \frac{\log \mathbb{E}_{\boldsymbol{c}} p(z,y|\boldsymbol{c})}{\log p(z,y|\boldsymbol{c})} \right) dz \, dy \geq 0. \tag{18}$$

The equality is achieved when the interaction combination $\boldsymbol{c}$ follows a specific distribution, such as $\boldsymbol{c}$ taking on a certain value with a predetermined probability. This aligns with our analysis of the data assumptions in Section 3.3. Thus, we can have the following conclusion:

$$I(Z;Y) - \mathbb{E}_{\boldsymbol{c}} I(Z;Y|\boldsymbol{c}) \geq \int p(z,y,\boldsymbol{c}) \log \frac{p(z|\boldsymbol{c})p(y|\boldsymbol{c})}{p(z)p(y)} \, dz \, dy \, d\boldsymbol{c} \tag{19}$$

And the term $\zeta = \mathbb{E}_{z,y,\boldsymbol{c}} \frac{p(z|\boldsymbol{c})p(y|\boldsymbol{c})}{p(z)p(y)}$ is irrelevant from the relationship between $z$ and $y$, thus it is model-agnostic. Hence, the high quality of the multimodal model can be determined when the information over different interactions is well learned. □

### A.2  PROOF FOR PROPOSITION 3.2

**Proposition A.2.** *Let $g$ be an estimator that maps samples $z$ and target $y$ to their pointwise mutual information. For a training set $S$ with interaction $c$, the estimator learned is denoted as $g_S$. Assume that for all $g \in \mathcal{G}, z \in \mathcal{Z}, y \in \mathcal{Y}$, the value $g(z,y)$ is bounded within $[-B, B]$. For any $\delta > 0$, with at least a probability of $1 - \delta$, the following inequality holds for each $g_S$:*

$$|I(Z;Y|\boldsymbol{c}) - I_S(Z;Y|\boldsymbol{c})| \leq 2\mathfrak{R}_S(\mathcal{G}) + B\sqrt{\frac{\log(1/\delta)}{2N}}, \tag{20}$$

*where $\mathfrak{R}_S(\mathcal{G})$ denotes the empirical Rademacher complexity, which measures the richness of the hypothesis class, and $N$ is the number of samples trained.*

*Proof.* First, we define the mutual information over the training phase and on the distribution, as:

$$\Phi(S) = \sup_{g \in \mathcal{G}} (\mathbb{E}[g] - \mathbb{E}_S[g_S]) = \sup_{g \in \mathcal{G}} (\mathbb{E}_{z,y}[g(z,y)] - \mathbb{E}_{x,y \in S}[g_S(z,y)]). \tag{21}$$

Let $S$ and $S'$ be two samples differing by exactly one point. Without loss of generality, let it be the last point, say $(z_N, y_N)$ in $S$ and $(z'_N, y'_N)$ in $S'$. Then, since the difference of suprema does not exceed the supremum of the difference, we have:

$$\Phi(S') - \Phi(S) \leq \sup_{g \in \mathcal{G}} (\mathbb{E}_S[g] - \mathbb{E}_{S'}[g]) = \sup_{g \in \mathcal{G}} \frac{g(z_N, y_N) - g(z'_N, y'_N)}{N} \leq \frac{B}{N} \tag{22}$$

When an $(z_i, y_i)$ pair changes, the random variable $\sup_{g \in \mathcal{G}} (\mathbb{E}_S[g] - \mathbb{E}_{S'}[g])$ can change by no more than $1/N$. McDiarmid's inequality implies that with probability at least $1 - \delta$,

$$\Phi(S) \leq \mathbb{E}_S[\Phi(S)] + B\sqrt{\frac{\log 1/\delta}{2N}} \tag{23}$$

We next bound the expectation of the right-hand side as follows

$$
\begin{aligned}
\mathbb{E}_S[\Phi(S)] &= \mathbb{E}_S \left[ \sup_{g \in \mathcal{G}} (\mathbb{E}[g] - \mathbb{E}_S(g)) \right] \\
&= \mathbb{E}_S \left[ \sup_{g \in \mathcal{G}} \mathbb{E}_{S'} [\mathbb{E}_{S'}(g) - \mathbb{E}_S(g)] \right] \\
&\leq \mathbb{E}_{S,S'} \left[ \sup_{g \in \mathcal{G}} (\mathbb{E}_{S'}(g) - \mathbb{E}_S(g)) \right] \\
&= \mathbb{E}_{S,S'} \left[ \sup_{g \in \mathcal{G}} \frac{1}{N} \sum_{i=1}^{N} (g(x_i') - g(x_i)) \right] \\
&= \mathbb{E}_{\boldsymbol{\sigma},S,S'} \left[ \sup_{g \in \mathcal{G}} \frac{1}{N} \sum_{i=1}^{N} \sigma_i (g(x_i') - g(x_i)) \right] \\
&\leq \mathbb{E}_{\boldsymbol{\sigma},S'} \left[ \sup_{g \in \mathcal{G}} \frac{1}{N} \sum_{i=1}^{N} \sigma_i g(x_i') \right] + \mathbb{E}_{\boldsymbol{\sigma},S} \left[ \sup_{g \in \mathcal{G}} \frac{1}{N} \sum_{i=1}^{N} -\sigma_i g(x_i) \right] \\
&= 2 \mathbb{E}_{\boldsymbol{\sigma},S} \left[ \sup_{g \in \mathcal{G}} \frac{1}{N} \sum_{i=1}^{N} \sigma_i g(x_i) \right] = 2\mathfrak{R}_S(\mathcal{G}).
\end{aligned}
\tag{24}
$$

Thus, we have completed the proof of the proposition. $\qquad \square$

## A.3 PROOF FOR LEMMA 3.3

**Lemma A.3.** *With the same Rademacher complexity, the main difference between multimodal joint learning and modality ensemble lies in the scale of samples, with the ensemble providing a tighter lower bound.*

*Proof.* For joint learning methods, the number of trained samples $N$ in Equation 8 is the same as the number of training samples $n$. However, considering modality ensemble with data with the redundancy interaction, each modality can learn the information contributing to the final result. We construct new data as $(\tilde{x}^{(1)}, \tilde{x}^{(2)}, y) \in \tilde{S} = \tilde{S}^{(1)} \cup \tilde{S}^{(2)}$, where $\tilde{S}^{(1)} = \{X^{(1)}, \Phi^{(2)}, Y\}$ and $\tilde{S}^{(2)} = \{\Phi^{(1)}, X^{(2)}, Y\}$, $\Phi^{(1)}, \Phi^{(2)}$ stay constant over each samples. The following inequality holds:

$$I(X; Y) = I(\{X^{(1)}, X^{(2)}\}; Y) \geq \max \left( I(X^{(1)}; Y), I(X^{(2)}; Y) \right). \tag{25}$$

And we will have:

$$H_S(Y) \geq I_{\tilde{S}^{(m)}}^{red}(\tilde{X}; Y) \geq \max \left( I_S^{red}(X^{(1)}; Y), I_S^{red}(X^{(2)}; Y) \right) = H_S(Y), \; m \in [2]. \tag{26}$$

Thus, the number of samples for modality ensemble is twice that of the dataset, satisfying $N = 2n$. Consequently, modality ensemble can effectively accomplish multimodal learning under redundancy interactions, resulting in a smaller lower bound on generalization. $\qquad \square$

## A.4 ANALYSIS FOR EQUATION 12

In this section, we analyze the correctness of the following inequality:

$$|I^{syn}(Z; Y) - I_S^{syn}(Z; Y)| \geq \max \left( I_S^{syn}(Z^{(1)}; Y), I_S^{syn}(Z^{(2)}; Y) \right). \tag{27}$$

Recall that the synergy interactions are defined as:

$$I^{syn}(X^{(1)}, X^{(2)}; Y) = H(Y), \quad I^{syn}(X^{(1)}; Y) = I^{syn}(X^{(2)}; Y) = 0. \tag{28}$$

Following the Data processing Inequality in Equation 2, we have the following inequality.

$$\begin{aligned} I^{syn}(Z^{(1)}; Y) &\leq I^{syn}(X^{(1)}; Y) = 0, \\ I^{syn}(Z^{(2)}; Y) &\leq I^{syn}(X^{(2)}; Y) = 0. \end{aligned} \tag{29}$$

Thus, features $z^{(m)}$ and the target $y$ are independent. We derive deeper in modeling their distribution, facilitating the analysis of the modality ensemble. We denote $p$ as the distribution under synergy interactions and we can denote the distribution as:

$$p(z^{(1)}, y) = p(z^{(1)})p(y), \ \ p(z^{(2)}, y) = p(z^{(2)})p(y). \tag{30}$$

Then we examine the modality ensemble with the weighted logits space. Here we define $f^{(1)}, f^{(2)} \in \mathbb{R}^k$ denote the unimodal logits, where $k = |Y|$, satisfying $p(y^1|x) = Softmax(f^{(1)})$, and $p(y^2|x) = Softmax(f^{(2)})$. The ensemble method is denoted as:

$$f = \alpha f^{(1)} + (1 - \alpha)f^{(2)} \tag{31}$$

Hence, under this kind of ensemble, we can obtain the distribution of the multimodal model:

$$p(y|z^{(1)}, z^{(2)}) = \frac{p(y|z^{(1)})^\alpha p(y|z^{(2)})^{1-\alpha}}{\sum_{j=1}^k p(y = j|z^{(1)})^\alpha p(y = j|z^{(2)})^{1-\alpha}} \tag{32}$$

which is obtained from the intrinsic nature of Redundancy. Bringing Equation 30 into Equation 32, we have that:

$$p(y|z^{(1)}, z^{(2)}) = \frac{p(y)^{\alpha+1-\alpha}}{\sum_{j=1}^k p(y = j)^{\alpha+1-\alpha}} = p(y) \tag{33}$$

Thus, we can conclude that :

$$I^{syn}(Z; Y) = H(Y) - I^{syn}(Y|Z) = \mathbb{E}_{z,y} \log \frac{p(y|z^{(1)}, z^{(2)})}{p(y)} = 0 \tag{34}$$

## A.5 EXPLANATION OF DECOMPOSITION

In this way, we investigate deeper to explain why our designed decomposition can effectively work. *w.l.o.g.*, we consider the process of decomposing the feature $Z$ the into two independent feature $V, T$. We introduce the variation distribution $q$, to help better model the likelihood:

$$\begin{aligned} \log p(z) &= \log \int p(z, v, t) \ dvdt = \log \int p(z|v, t)p(v)p(t) \ dvdt \\ &\geq E_{q(v|z), q(t|z)} \log p(z|v, t) \frac{p(v)}{q(v|z)} \frac{p(t)}{q(t|z)} \\ &= E_{q(v|z), q(t|z)} \log p(z|v, t) + E_{q(v|z)} \frac{p(v)}{q(v|z)} + E_{q(t|z)} \frac{p(t)}{q(t|z)} \\ &= E_{q(v|z), q(t|z)} \log p(z|v, t) - KL(q(v|z)||p(v)) - KL(q(t|z)||p(t)). \end{aligned} \tag{35}$$

Hence, we can enhance the ELBo to help estimate the distribution of the probability of $p(z)$. From another perspective, this decomposition can be represented using mutual information, aiming that:

$$\min I(Z; T) + I(Z; V) - I(Z; T, V) = I(Z; T; V) = I(T; V) \tag{36}$$

The second equation is based on the assumption that $p(v|t, z) = p(v|z)$, that is given $z$, $v, t$ are independent. Then this objective aligns with the ELBo Equation 35, that is:

$$I(Z; T, V) = \int p(z, t, v) \log \frac{p(z|v, t)}{p(z)} \ dvdtdz \leq E_{p(z)} E_{q(v|z), q(t|z)} \log p(z|v, t) - H(Z);$$

$$I(Z; T) = \int p(z, t) \log \frac{p(t|z)}{p(t)} \leq E_{p(z)} E_{q(t|z)} \frac{q(t|z)}{p(t)} = E_{p(z)} KL(q(t|z)||p(t)), \tag{37}$$

It is obvious that the minimize the upper bound Equation 37 is similar to the maximize the ELBo.

| Dataset | UR-FUNNY (V+T) | | ROSMAP (mRNA+METH) | | VGGsound (A+V) | |
|---|---|---|---|---|---|---|
| Metric | ACC | F1 | ACC | F1 | ACC | F1 |
| Joint | 63.8 | 63.7 | 84.0 | 83.8 | 55.1 | 53.3 |
| Ensemble | 63.2 | 63.2 | 83.0 | 83.0 | 56.7 | 55.1 |
| DMI | 65.0 | 64.7 | 84.9 | 84.9 | 58.5 | 57.0 |

Table 5: Validating on diverse modalities and various scales.

## B    EXPERIMENTAL DETAILS

### B.1    EXPERIMENTAL SETTING

**Training Details**    The training process used a batch size of 64 for CNN-based methods and 32 for Transformer-based methods. The learning rate was specifically set for each dataset, ranging from 1e-2 to 1e-3. We employed SGD as the optimizer, with a momentum of 0.9 and a weight decay of 1e-4. In our method, the transition from stage 1 to stage 2 occurred around epoch 5, and the switch from stage 2 to stage 3 occurred around epoch 10.

**Data-preprocessing**    Videos in Kinetics-Sounds last 10 seconds in length and we extract frames with 1 fps. Considering the difference between datasets, 3 frames are uniformly sampled from each 10-second clip as visual inputs. For CREMA-D, we extract 1 frame from each of the clips. For UCF101, we choose 2 frames RGB and 5 frames of optical flow. For VGGsounds, we choose 3 frames visual inputs.

### B.2    MODEL ARCHITECTURE

In this section, we elaborate the architecture of our Decomposition-based Multimodal Interaction learning (DMI) model. A comprehensive illustration is presented in Figure 3. The DMI model consists of two distinct decomposition modules. Initially, samples from $m$ modalities, denoted as $X^{(m)}$, are encoded into features $Z^{(m)}$ through modality-specific encoders $\phi^{(m)}$. These features are then decomposed to elucidate intermodal interactions. Each decomposition module is structured on a Variational Autoencoder (VAE) framework, where the encoders, composed of Multi-Layer Perceptrons (MLPs), predict the mean and variance. Conversely, the decoders are designed as multilayer networks to ensure minimal information loss during the decomposition process. The alignment of features across modalities is enforced by minimizing the Kullback-Leibler (KL) divergence between the corresponding distributions.

For effective training, we not only minimize the decomposition loss but also focus on task-related feature decomposition. This requires the task-related feature $T$ to encapsulate all necessary information pertaining to the specific task at hand. Consequently, additional loss functions are integrated to ensure that the feature $T$ effectively contributes to the final task performance.

### B.3    EXPANDED EXPERIMENTS ON VARIOUS DATASETS

### B.3.1    DATASETS

**ROSMAP** (De Jager et al., 2018): This dataset is used for Alzheimer's Disease diagnosis and includes mRNA and METH modalities. It contains 351 samples across 2 classes. **UR-FUNNY** Hasan et al. (2019): This first proposed large-scale multimodal dataset for humor detection combines text, visual, and acoustic modalities. It comprises over 16,000 video samples from TED talks, showcasing a variety of speakers and topics. This diversity makes it ideal for modeling multimodal language and understanding humor. **VGGsound** Chen et al. (2020): This audio-visual dataset consists of short clips from over 200,000 YouTube videos, capturing sounds in diverse acoustic environments.

| Dataset | CMU-MOSEI (V+A+T) | | UCF (RGB+OF+Diff) | |
|---|---|---|---|---|
| Metric | ACC | F1 | ACC | F1 |
| Joint | 63.3 | 63.2 | 78.6 | 78.2 |
| Ensemble | 63.4 | 62.7 | 84.4 | 83.9 |
| DMI-TD | **64.3** | **64.5** | **84.8** | **84.2** |

Table 6: Experiments on CMU-MOSEI and UCF datasets with three modalities.

| Temporal | CREMA-D-2Frame | | CREMA-D-8Frame | | KS-8Frame | |
|---|---|---|---|---|---|---|
| Metric | ACC | F1 | ACC | F1 | ACC | F1 |
| Joint | 77.8 | 78.3 | 85.5 | 85.9 | 85.3 | 85.3 |
| Ensemble | 77.7 | 78.2 | 86.6 | 87.0 | 87.1 | 87.1 |
| DMI | **78.5** | **79.3** | **87.5** | **87.9** | **87.5** | **87.5** |

Table 7: Experiment validating on richer temporal dynamics.

### B.3.2 EXTENSIVE MODALITY AND TASKS

To get holistic validation of our method, we introduce three additional datasets: UR-FUNNY, ROSMAP, and VGGSound. These datasets were chosen to address new tasks, modalities, and larger-scale data. UR-FUNNY is a humor detection dataset, where we use Visual (V) and Text (T) modalities. These two modalities are considered to contain significant synergy information (Liang et al., 2023b). ROSMAP is a biological dataset that introduces new modalities: mRNA and methamphetamine (METH). VGGSound is a large-scale dataset containing over 200,000 samples.

The experimental results are shown in Table 5. We observe that ensemble methods perform worse than joint learning on the humor and biology datasets. Since UR-FUNNY contains numerous synergy interactions, which are difficult to capture using modality ensembles (Table 1), our method significantly outperforms ensemble models. This highlights the ability of our method to learn multimodal interactions adaptively. Similarly, our method performs well on biological modalities and large-scale data, further demonstrating its effectiveness across a variety of scenarios.

### B.3.3 EXTENDED TO THREE MODALITIES

The presence of diverse modalities in real-world scenarios poses significant challenges for multimodal methods, particularly concerning their extension to more than two modalities. We consider this extension from both analytical and experimental perspectives.

Analytically, the interactions among more than two modalities introduce substantial complexity. Defining mutual information across four variables, for instance, presents significant theoretical challenges, especially when attempting to characterize concepts such as redundancy, uniqueness, and synergy Liang et al. (2023b). However, experimentally, our proposed Decomposition-based Multimodal Interaction learning (DMI) approach can be modified to adapt to scenarios involving three modalities. Specifically, by implementing Task-related Decomposition on DMI (DMI-TD, as illustrated in Figure 4), our paradigm can accommodate three modalities.

Empirical evaluations were conducted on two datasets incorporating three modalities each: CMU-MOSEI, which includes Visual (V), Audio (A), and Text (T) modalities, and UCF101, consisting of RGB, Optical Flow (OF), and Frame Difference (Diff) modalities. The experimental results, as shown in Table 6, highlight the enhancement achieved by our DMI method, thus showcasing the flexibility and effectiveness of our approach.

### B.3.4 VALIDATED ON RICHER TEMPORAL DYNAMICS

Previous literature underscores the importance of temporal dynamics in enhancing multimodal tasks (Bernin et al., 2018). Consequently, examining the influence of richer temporal dynamics on model performance presents a pertinent research question. We select the CREMA-D and KS datasets to val-

| Dataset | CMU-MOSEI | | KS | |
|---------|------|------|------|------|
| Backbone | LSTM | | ResNet34 | |
| Metric | ACC | F1 | ACC | F1 |
| Joint | 62.4 | 62.2 | 86.0 | 85.8 |
| Ensemble | 62.0 | 61.7 | 86.8 | 86.3 |
| DMI | **62.9** | **62.9** | **87.8** | **87.7** |

Table 8: Validation across different backbone on CMU-MOSEI (V+T) and KS (A+V) datasets.

idate our method under richer temporal information. For CREMA-D, we used 2 and 8 frames, while for KS, we extracted 8 frames. The experimental results are shown in Table 7. When compared to the results in Table 2, CREMA-D shows a significant improvement with more temporal information, whereas KS exhibits a more modest enhancement. Moreover, we observed that, on the CREMA-D dataset, ensemble learning improves more than the joint learning method with the number of frames increasing. This suggests that enhanced temporal dynamics elevate the unimodal information, making it easier for the ensemble method to utilize this information effectively. Additionally, our proposed DMI method still outperforms these baselines, further confirming its effectiveness.

### B.3.5 VARIOUS BACKBONE

Different backbone architectures process data uniquely, thereby capturing and learning interactions in distinct ways. In previous experiments in Table 2 and Table 3, we evaluated CNN-based and Transformer-based backbones. To further explore the impact of different architectures, we have implemented additional validation by changing the backbone and validating the CMU-MOSEI and KS datasets. Specifically, for the CMU-MOSEI dataset, we employed an LSTM as the backbone (Liang et al., 2021) on Visual and Text modalities, while for the KS dataset, we utilized ResNet34 as the backbone. The experimental results are shown in Table 8. These results suggest that changing the backbone can influence baseline performance. Specifically, the joint model marginally underperforms the ensemble model when using a Transformer backbone, but outperforms it on the LSTM backbone. By combining the findings from Table 8 with those in Table 2 and Table 3, we further demonstrate that our method not only outperforms other approaches but also effectively learns interactions across different backbones.

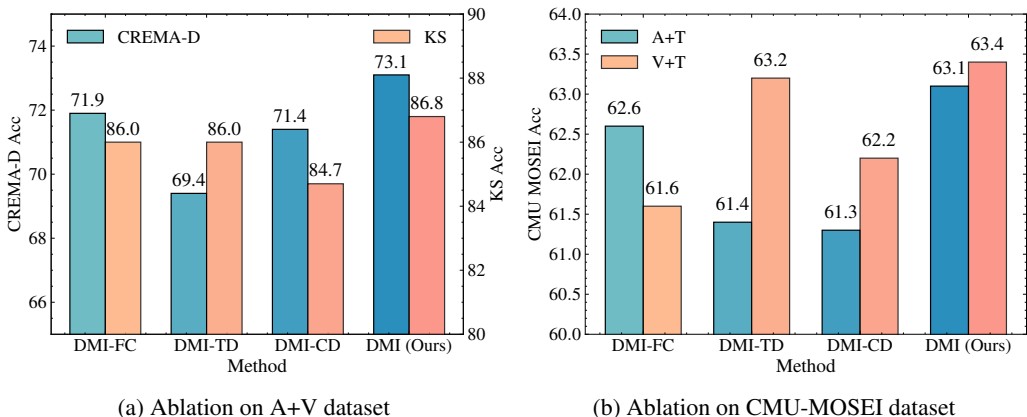

(a) Ablation on A+V dataset    (b) Ablation on CMU-MOSEI dataset

Figure 5: Ablation studies of DMI paradigm on CREMA-D, Kinetics Sounds(KS) and CMU-MOSEI datasets with Audio+Text and Visual+Text modalities.

### B.3.6 EXTENSIVE ABLATION STUDY

We add some ablation studies for further analysis. Combin. The study addresses two primary questions: *1) Is the use of variational methods indispensable? 2) Are both decomposition modules essential for the framework's performance?* In the experiments, we introduce *DMI-Fully Connected*

(DMI-FC), which realizes decomposition using fully connected layers instead of variational information bottlenecks. Additionally, we explore configurations that omit either the *Task-related Decomposition* (TD) module or the *Consistent Decomposition* (CD) module. Specifically, DMI-TD retains only the TD module, while DMI-CD preserves the CD module, depicted in Figure 5.

On one hand, we observe that the performance of DMI-FC, which incorporates the full decomposition process, is higher than that of the other two ablation settings—DMI-TD and DMI-CD, which only retain partial decomposition modules. However, in more challenging tasks, such as the (V+T) setup in CMU-MOSEI, DMI-FC's performance can deteriorate significantly. This issue is mitigated by employing variational decomposition, which effectively decouples different interactions and handles challenging datasets. On the other hand, while DMI-TD and DMI-CD, each employing a specific decomposition method, can improve performance to some extent, the overall DMI—combining both decomposition modules—consistently achieves superior and stable improvements. This underscores the importance of employing both decomposition networks. These findings highlight the critical role of variational interaction decomposition modules within our framework.

### B.4 SYNTHETIC DATASET

In the Section 4.3, we construct two types of synthetic data to facilitate our study. The first type is shown in Figure 2, where the data is crafted with pre-defined interactions to elucidate specific interaction dynamics. The second type, depicted in Table 4, derives from Boolean logic variables. Here, the interactions are inherently embedded within the Boolean logic itself, providing a general consideration for interaction analysis.

The data generation process for predefined interactions is executed in two sequential steps. Initially, the type of interaction for each sample is identified. We categorize potential interactions as *R*edundancy, *U*niqueness, and *S*ynergy for each dataset. As illustrated in Figure 2, each dataset is composed of samples exhibiting one or two types of interactions. The proportion of each interaction type is quantified using a fractional notation, such as $\frac{1}{4}U + \frac{3}{4}R$. This indicates that $\frac{1}{4}$ of the samples display **Unique** interactions, while the remaining samples demonstrate **Redundant** interactions.

In the second step, data corresponding to the predefined interactions are constructed. Different networks are employed to encode specific interactions into some dimensions of input space, which are then concatenated to form a comprehensive sample. If a sample is defined as a certain interaction, other types of interaction are suppressed by introducing Gaussian noise into their respective dimensions. This approach ensures that each sample exclusively embodies one type of interaction, thereby facilitating the construction of datasets with specified interaction properties.

The dataset, derived from Boolean logical data, is generated in a structured manner. Initially, the specific Boolean logic within each sample is determined. We consider two to three types of Boolean logics—*AND*, *OR*, and *XOR*—with each sample containing only one type. Each type of logic occupies an equivalent proportion within the dataset. Subsequently, these logics are encoded into the input space. Given that Boolean data inherently contain measurable interactions Bertschinger et al. (2014), we utilize this data to validate our method for interaction decomposition Table 4.

