# OpenReview forum: "Towards Holistic Multimodal Interaction: An Information-Theoretic Perspective"
_ICLR.cc/2025/Conference — Submitted to ICLR 2025_

### Official Review · Reviewer_9UkZ · 2024-10-18

**Soundness:** 3
**Presentation:** 3
**Contribution:** 2
**Rating:** 6
**Confidence:** 5

**Summary:**

The paper investigates the role of multimodal interaction in multimodal learning. It provides information-theoretical evidence that learning all types of interactions (redundancy, uniqueness, synergy) is necessary for good performance and shows that naive joint and ensemble learning cannot learn all types of interactions equally well. Motivated by this finding, the paper proposed the Decomposition-based Multimodal Interaction learning (DMI) paradigm that uses variation-based approach to decompose multimodal information into different types of interactions learned via a three-phase training.

**Strengths:**

1. **Novel theoretical analysis**: the paper presents an original theoretical analysis on the role of multimodal interaction in multimodal learning, being the first to establish a theoretical connection between multimodal interaction and multimodal learning performance. The analysis is clear, well-written, accompanied by sound proofs, which provides theoretical insights to understanding the importance of multimodal interaction in addition to prior works. The paper also proves that naive joint and ensemble learning are not able to capture all types of necessary interactions, resulting in suboptimal performance and generalization gap.
2. **New learning paradigm for interaction learning**: the paper proposes a new learning paradigm, DMI, designed to explicitly disentangle and capture the three types of interactions and corresponding training strategy for DMI learning. Evaluation on real-world datasets corroborate some claims of effectiveness of this learning paradigm.

**Weaknesses:**

1. **Generally weaker experiment section**: the paper perform a relatively comprehensive evaluation of relevant methods that also target at capturing interactions in multimodal learning; however, the analysis on the results is generally very limited.
- There is no comparison/analysis between the proposed DMI and regulation methods and other interaction methods (the paper only mentions that the improvement attributes to the effective decomposition and holistic learning of different interactions, which is weakly supported by ablations, further analysis, or the additional experiment details in appendix). Meanwhile, the reviewer is not sure whether it could be a fair comparison (e.g. are modality-specific encoders and model size standardized?)
- The improvement from existing best-performing methods / baselines seems marginal ($\le$ 2% in accuracy), which could undermine the claims about importance of learning interactions in multimodal learning.
- The choice of evaluation datasets / benchmarks are not justified and are also limited in terms of scope, given a wide range of multimodal datasets / benchmarks exist.

2. **Documentation of the synthetic setup needs more details**: the section on validating DMI on synthetic data and Figure 1 are interesting, but the documentation of data generation and experimental setup need more details. For example, the following aspects need more clarifications:
- How do the informative dimensions "maintain statistical relationships with the label" specifically? What are the statistical relationships that represent the three types of interactions respectively?
- How is different interaction combination (e.g. $\frac14 U+\frac34 R$, Figure 1) achieved in the data generation?
- How many synthetic data are used in this validation setting? What is the DMI architecture evaluated in this setting?
- Comparing to CVX, DMI indeed show better approximation but the evaluation is limited to one complex setting AND+XOR. Maybe adding a few evaluations on other complex logical relations (e.g. OR+XOR, AND+XOR+OR) could strengthen the claim.

In general, the reviewer agrees that if the paper is primarily a theoretical contribution, it does not need to incorporate evaluations on real-world benchmarks as comprehensively as other empirical studies, but the reviewer also believes it is necessary to document the details of data generation and experimental setup of the presented results in the appendix, which is currently missing. The reviewer believes that increasing the completion and soundness of the experiments and analysis section (compared to the theoretical section) will make this submission much stronger.

**Questions:**

General questions / Clarifications:
- Major questions mentioned in the Weaknesses section, including more details on experimental setup for fair comparison across methods, justification / limitation in the choice of evaluation datasets, more details on synthetic data generation and more validation settings
- Can the author clarify why synergistic information are considered as task-irrelevant features? Synergy can be very important for pure-synergy tasks such as XOR.
- **Concerns about DMI assumption**: DMI assumes a complete separation of synergistic features and features of other interactions (redundancy, uniqueness). However, can overlaps in features exist? For example, in detecting sarcasm, if a person is criticizing with a smiley face, the positive sentiment in facial expression can be unique in the visual modality, while the negative sentiment in speech is only available in the text modality, but both these features are also necessary for the inconsistency, i.e. the synergistic information, which is essential for sarcasm detection. Is DMI able to handle such cases, and how does it perform on sarcasm detection?

Notations:
- A.1 Proof for Proposition 3.1: the first term should be $\log\frac{\mathbb{E}_\boldsymbol{c}p(z,y|\boldsymbol{c})}{p(z,y|\boldsymbol{c})}$ instead of $\frac{\log\mathbb{E}_cp(z,y|\boldsymbol{c})}{\log p(z,y|\boldsymbol{c})}$ in equation 17, 18
- A.5 Explanation of decomposition: inconsistent notation $u,v$ for the first independent features

---

> ### Author Response · Authors · 2024-11-26
> **Response by authors**
>
> **Question 1. About real-world experiments:**
>
> > Question 1a. lack extensive empirical analysis.
>
> Thank you for your valuable comment. At present, measuring interactions under real data conditions remains imprecise. This challenge is largely due to the lack of an accurate understanding of interactions of each sample, which remains an open problem in the field. To address this, we have taken two approaches. First, we expanded our experimental scope significantly to demonstrate the versatility of our method across various contexts, which is detailed in Appendix B.3. The validation experiments including extensive modalities, tasks (Appendix B.3.2), and backbones (Appendix B.3.5), three modalities (Appendix B.3.3), with temporal dynamics (Appendix B.3.4) and more analysis about ablation (Appendix B.3.6). Second, we conducted validation experiments on synthetic datasets with controllable interactions for each sample to more clearly illustrate the learning mechanisms of our method. Moving forward, with more accurate measurements of multimodal interactions to be investigated, we expect to gain a deeper understanding of our approach.
>
>
> > Question 1b. Marginal improvement over some datasets.
>
> Thank you for raising this point. These results were observed in the AV-MNIST and CMU-MOSEI (V+T) task, which is inherently challenging.
> AV-MNIST, a synthetic dataset derived from MNIST, has traditionally seen only modest improvements with previous methods compared to joint training. For example, the best performance in prior work is 72.8%, compared to 71.7% for joint training [1]. In our experiments, our method achieved 73.5%, significantly surpassing the joint training benchmark of 72.6%, thereby validating the efficacy of our approach. Additionally, we have replaced this synthetic dataset with real-world data from the UCF-101 dataset, which includes Optical Flow and RGB modalities, as detailed in Table 2. We present partial results here, demonstrating the effectiveness of our method across real-world datasets.
>
> |        |     | RGB   | OF    | Joint | Ensemble | DMI   |
> |--------|-----|-------|-------|-------|----------|-------|
> | UCF101 | ACC | 76.9  | 67.8  | 78.8  | 82.3     | **84.2**  |
> |        |  F1 | 76.1  | 67.6  | 78.0  | 81.8     | **83.9**  |
>
> Sentiment analysis tasks are particularly challenging on the CMU-MOSEI dataset, where even marginal improvements are valuable [2]. Previous studies have reported a maximum improvement of 1.5% over joint learning (LF-Transformer) in binary classification tasks across three modalities [1]. In this paper, we tackle a more challenging setup with only two modalities, making the task even more difficult. Our method achieves a 1.2% increase in the (A+T) setup for a three-way classification task. Moreover, our method demonstrates improvement in the (V+T) modality, whereas some comparison methods experience a drop in performance. This highlights both the complexity of learning in this setup and the effectiveness of our proposed approach.
>
> > Question 1c. Dataset, benchmarks are also limited in terms of scope.
>
> Thank you for your suggestion. Existing datasets typically include Audio, Visual, and Text modalities, with sample sizes ranging from 5,000 to 30,000. Following your suggestion, we have expanded our experiments to include more modalities and larger-scale datasets.
> In the revised manuscript, we validate our methods across various datasets and combinations of modalities:  Audio + Text on the UR-FUNNY dataset for humor detection [3], RGB + Optical Flow on the UCF101 dataset for action recognition[4], mRNA + methylation data on the ROSMAP dataset for Alzheimer's Disease diagnosis [5], and Audio + Visual on the VGGsound dataset for audio recognition, which includes an extensive sample size of 200,000 [6]. Detailed results for these diverse modalities are presented in Appendix B.3. Experimental outcomes confirm that our method significantly outperforms the baseline across different scales and modalities.
>
> |  Dataset | UR-FUNNY|       | UCF101 |       | ROSMAP  |       | VGGsound |       |
> |:--------:|:--------------:|:-----:|:------------:|:-----:|:------------------:|:-----:|:--------------:|:-----:|
> |  Method  |       ACC      |   F1  |      ACC     |   F1  |         ACC        |   F1  |       ACC      |   F1  |
> |   Joint  |      63.8      | 63.7  |     78.8     | 78.0  |        84.0        | 83.8  |      55.1      | 53.3  |
> | Ensemble |      63.2      | 63.2  |     82.3     | 81.8  |        83.0        | 83.0  |      56.7      | 55.1  |
> |    DMI   |      **65.0**      | **64.7**  |     **84.2**     | **83.9**  |        **84.9**        | **84.9**  |      **58.5**      | **57.0**  |

---

> ### Author Response · Authors · 2024-11-26
> **Response by authors**
>
> **Question 2. About synthetic experiment:**
>
> > Question 2a. The statistical relationships of three types of interactions
>
> Thank you for highlighting this point. For synthetic data, we specifically construct samples to correspond to particular types of interactions. Each sample exclusively contains one type of interaction, and for simplicity, different interactions are encoded in distinct dimensions of the inputs. When a sample is determined by specific interaction, dimensions corresponding to other interactions will be set into noise. This approach ensures that each sample uniquely represents one type of interaction. We provide further clarification of this setting in Appendix B.4.
>
> > Question 2b. Interaction generation for $\frac{1}{4} U + \frac{3}{4} R$.
>
> Thank you for raising this point. In Figure 1, two types of interactions are depicted: redundancy and uniqueness. The notation $\frac{1}{4} U + \frac{3}{4} R$ indicates that $ \frac{1}{4} $ of the data is sampled from unique interactions, while $ \frac{3}{4} $ is sampled from redundant interactions. Each sample corresponds with a certain type of interaction. We clarify this description in the revised manuscript, Appendix B.4.
>
> > Question 2c. Synthetic data and DMI architecture.
>
> Thank you for your question. Here, we use 10000 samples in the synthetic dataset, and the DMI architecture is similar to the real-world experiment (detailed in Appendix B.2), with the backbone changing into a 3-layer neural network with ReLU as an activate function.
>
> > Question 2d. More boolean mixture of interactions.
>
> Thank you for pointing this out. In response, we have incorporated additional types of interactions, including combinations of (OR + XOR) and (AND + OR + XOR), to assess the effectiveness of our method in handling more complex interactions. The detailed results indicate that the DMI approach can effectively capture these complex interactions, even when multiple Boolean variables are involved. These results are presented below and are also summarized in Table 4 of the revised manuscript.
>
> |        |       | OR+XOR |       |       |       | AND+OR+XOR |       |       |
> |--------|:-----:|:------:|:-----:|:-----:|:-----:|:----------:|:-----:|:-----:|
> | Method | $R$   | $U_1$  | $U_2$ | $S$   | $R$   | $U_1$      | $U_2$ | $S$   |
> | DMI    | 27.65 | 4.76   | 0     | 67.59 | 20.79 | 0          | 2.36  | 76.85 |
> | CVX    | 33.66 | 0.37   | 0.14  | 65.83 | 21.16 | 0          | 0.58  | 78.26 |
> | Truth  | 25.51 | 0      | 0     | 74.49 | 19.1  | 0          | 0     | 80.9  |
>
> **Question 3. Why synergistic information are considered task-irrelevant features?**
>
> Thank you for addressing this important aspect. Task-irrelevant information is defined as **information within unimodality** that is **not directly** related to the task at hand. According to the definition of synergy data (refer to Equation 11 and [1]), synergy arises when **individual modalities alone provide no information for task completion**, yet their integration generates emergent information that is crucial for the task. This emergent information, which we term synergy, arises indirectly from unimodality. Consequently, the information derived from synergy interaction belongs to the task-irrelevant category within each unimodality. Thus, we characterize the information emergent from the combination of two task-irrelevant features as the learned synergy. We clarify this distinction in Section 3.4.

---

> ### Author Response · Authors · 2024-11-26
> **Response by authors**
>
> **Question 4. Does overlap lie in DMI assumption?**
>
> Thank you for this invaluable question. To illustrate this point, let us consider the use of a smiley face in sarcasm detection within the visual modality. The interpretation of the smiley face as conveying unique information is highly context-dependent. For example, if every data point includes a smiley face, there is no discernible correlation between its presence and sarcasm—actually, the smiley face conveys no information about sarcasm in this scenario. Conversely, if the smiley face appears exclusively under specific conditions, such as during moments of happiness or sarcasm, it then strongly correlates with sarcasm, thus illustrating the unique interaction our method aims to capture.
>
> Our approach is designed to decompose each data point into multiple types of interactions. In the scenario described, the uniqueness interaction captures task-related correlations, while the synergy interaction identifies emergent correlations.
>
> We have also applied our methodology to the humor detection dataset UR-FUNNY, which is known to contain an amount of synergy information [7]. The experimental results demonstrate that our method can effectively capture this synergy information, leading to improved performance.
>
> |  Dataset | UR-FUNNY (V+T) |       |
> |:--------:|:--------------:|:-----:|
> |  Method  |       ACC      |   F1  |
> |   Joint  |      63.8      | 63.7  |
> | Ensemble |      63.2      | 63.2  |
> |    DMI   |      65.0      | 64.7  |
>
>
> [1] P. P. Liang, Y. Lyu, X. Fan, Z. Wu, Y. Cheng, J. Wu, L. Chen, P. Wu, M. A. Lee, Y. Zhu et al., “Multibench: Multiscale benchmarks for multimodal representation learning,” *arXiv preprint arXiv:2107.07502*, 2021.
>
> [2] H. Wang, S. Luo, G. Hu, and J. Zhang, “Gradient-guided modality decoupling for missing-modality robustness,” in *Proceedings of the AAAI Conference on Artificial Intelligence*, vol. 38, no. 14, 2024, pp. 15 483– 15 491.
>
> [3] M. K. Hasan, W. Rahman, A. Zadeh, J. Zhong, M. I. Tanveer, L.-P. Morency et al., “Ur-funny: A multimodal language dataset for understanding humor,” *arXiv preprint arXiv:1904.06618*, 2019.
>
> [4] K. Soomro, “Ucf101: A dataset of 101 human actions classes from videos in the wild,” *arXiv preprint arXiv:1212.0402*, 2012.
>
> [5] P. L. De Jager, Y. Ma, C. McCabe, J. Xu, B. N. Vardarajan, D. Felsky, H.-U. Klein, C. C. White, M. A. Peters, B. Lodgson et al., “A multi-omic atlas of the human frontal cortex for aging and alzheimer’s disease research,” *Scientific data*, vol. 5, no. 1, pp. 1–13, 2018.
>
> [6] H. Chen, W. Xie, A. Vedaldi, and A. Zisserman, “Vggsound: A large-scale audio-visual dataset,” in *ICASSP 2020-2020 IEEE International Conference on Acoustics, Speech and Signal Processing* (ICASSP). IEEE, 2020, pp. 721–725.
>
> [7] P. P. Liang, Y. Cheng, X. Fan, C. K. Ling, S. Nie, R. Chen, Z. Deng, F. Mahmood, R. Salakhutdinov, and L.-P. Morency, “Quantifying & modeling multimodal interactions: An information decomposition framework,” in *Advances in Neural Information Processing Systems*, 2023.

---

> ### Author Response · Authors · 2024-11-29
> **Look forward to your feedback!**
>
> Dear Reviewer 9UkZ,
>
> Thank you for reviewing our work and for providing constructive suggestions. We have carefully considered your comments and made the necessary adjustments. We would be thankful if you could inform us whether the revisions adequately address your concerns. If further clarification is needed, we are happy to provide additional details.

---

> > ### Comment · Reviewer_9UkZ · 2024-12-01
> >
> > Thank you for the efforts in the additional experiments and clarifications. They have mostly addressed the questions I raised, especially strengthening the experiment section. I think with incorporating the changes, the submission now can be considered as a good contribution to the research in improving multimodal learning from a better understanding (theory) and capturing (the proposed framework) of multimodal interactions. Therefore, I have raised my rating to 6 (soundness to 3). My other ratings remain unchanged.

---

> ### Author Response · Authors · 2024-12-01
> **Thank you!**
>
> We greatly appreciate your positive feedback and are pleased to hear that your questions have been addressed.
>
> Thank you again for your valuable insights. If you require any further clarification or additional experiments, please feel free to reach out to us.

---

### Official Review · Reviewer_a4Ud · 2024-11-04

**Soundness:** 3
**Presentation:** 3
**Contribution:** 3
**Rating:** 6
**Confidence:** 4

**Summary:**

The paper introduces an information-theoretic framework that shows the importance of learning from different interactions and the shortcomings of the traditional multimodal methods. To solve the issue, the paper proposed a decomposition-based multimodal interaction learning model that disentangles the interactions within the multi-modal data. Experiments on the various datasets and straightforward visualizations show the effectiveness of the proposed method.

**Strengths:**

1. The visualizations are clear and straightforward. The figures are well-designed.
2. Theoretic analysis demonstrates the importance of learning from different interactions and the shortcomings of traditional multimodal methods.
3. The proposed method shows promising performance on most of the datasets and tasks against competitive SOTA baselines.

**Weaknesses:**

1. The method section lacks sufficient detail and thus is a bit confusing to me. Please refer to Questions 1-6.
2. DMI’s improvements over the best baseline on AV-MNIST and CMU-MOSEI (V+T) are not statistically significant.

**Questions:**

1. Line 322, why does synergy only contain task-irrelevant information? My understanding is that the integration of synergy has additional information to unimodal data which is task-relevant.
2. Equation 13, V and M are symmetric, which implies they can be interchanged without affecting the outcome. Given that, how do you control the learning to make one vector task-relevant and the other task-irrelevant? (which is mentioned in Line 352)
3. Line 368, training stage 1 lacks sufficient detail. Can you explain how you warm-up the encoder? Specifically, what are the input, output, and objective during this phase? Additionally, Figure 3 shows two encoders within distinct decomposition components. Are both encoders warmed up in this stage?
4. Line 371, training stage 2, “we freeze the encoder and focus solely on training the decomposition module…” Could you clarify if this means that all encoders are frozen, with only the decoders being fine-tuned in this stage?
5. In Figure 3, the decomposition modules include decoders. Does the learning objective (Equations 13 and 14) have a reconstruction loss to guide the decoders during training?
6. After decomposition is complete, how is the pre-trained model utilized for downstream tasks? Specifically, is there any further fine-tuning involved, or do you directly apply the representations learned from the decomposition modules to the downstream tasks?
7. Line 416,  can you explain the several modifications specific to each modality in ResNet18?
8. For MOSEI, are you working on sentiment analysis or emotion recognition?
9. Tables 2 and 3, performance of the uni-modal method is missing.
10. Section 4.4 ablation study, the study shows that using a single decomposition method (DMI-TC and DMI-CD) yields worse performance than the approach without any decomposition at all (DMI-FC). Can you explain the reason?

---

> ### Author Response · Authors · 2024-11-26
> **Response by authors**
>
> **Question 1. Question with the method section.**
>
> Thank you for your valuable comment. Here, we provide a detailed explanation of questions and modify the paper to describe our method detailedly.
>
> > Question 1a. Why does synergy only contain task-irrelevant information?
>
> Thank you for addressing this important aspect. Task-irrelevant information is defined as **information within unimodality** that is **not directly** related to the task at hand. For example, in XOR data with two independent binary variables, there is no correlation between the target and either modality. However, when two variables are not independent (e.g., $x^1 = x^2$), task-related information occurs, meanwhile additional interactions beyond synergy are observed.
> Thus, the information derived from synergy interactions falls into the task-irrelevant category within each unimodality. We characterize the information that emerges from the combination of two task-irrelevant features as the learned synergy. This distinction is further clarified in Section 3.4.
>
> > Question 1b. How to control the learning to make one vector task-relevant and the other task-irrelevant?
>
> Thank you for this inquiry. Our approach draws inspiration from the Variational Information Bottleneck (VIB) technique [2], where we utilize unimodal task-relevant features to finish specific tasks. To effectively separate task-relevant and task-irrelevant information, we incorporate an additional task-related loss function.
> As detailed in Equation 13, the VIB framework allows us to reduce the information flow between representation $Z^{(m)}$ into two decoupled components, task-related vector $T^{(m)}$ and task-irrelated vector $V^{(m)}$. By minimizing the mutual information terms, $I(Z^{(m)}; T^{(m)})$ and $I(Z^{(m)}; V^{(m)})$, we can reduce the consistent information between $T^{(m)}$ and $V^{(m)}$. Hence, the two vectors can be decoupled and represent task-relevant and task-irrelevant information, respectively.
>
> > Question 1c. How to warm-up the encoder.
>
> Thank you for your thoughtful comment. In our method, we specifically warm-up **the unimodal encoder**, which is crucial for the process of transforming $X^{(m)}$ into $Z^{(m)}$. It is important to note that **the variational encoder does not require warming up**, a distinction we have clarified in the revised manuscript.
> The rationale behind warming up the unimodal encoder stems from the observation that, in the early stages of learning, these encoders are not yet capable of extracting specific information effectively. Initiating the learning process without a warm-up phase often results in suboptimal performance due to premature decomposition.
> To address this, we train every modality respectively with a few epochs to the target, similar to the modality ensemble paradigm. This approach ensures that each modality’s encoder is adequately prepared to extract and subsequently decompose the information effectively.
>
> > Question 1d. Explanation of stage 2.
>
> Thank you for your question. In this stage, our objective is to refine the decomposition process to effectively distinguish between different types of interactions. To achieve this, we freeze the **unimodal encoder, $\phi^{(m)}$**, which allows us to focus solely on training the decomposition network without the interference of evolving unimodal representations.
> We have updated and clarified this process in Section 3.4.
>
> > Question 1e. About reconstruction objective.
>
> Thank you for your question. Indeed, we incorporate a reconstruction loss as part of our learning objective, in line with other VAE-based decomposition methods. This reconstruction loss is essential for minimizing the information loss that occurs after processing by the variational encoder. For a detailed explanation of our model, please refer to Appendix B.2 in the revised manuscript.
>
> > Question 1f. How to fine-tune.
>
> This is an excellent point. In this paper, we employ a straightforward yet effective method for fine-tuning: we directly integrate the interaction variables and use them to complete the specified task. Specifically, we concatenate these variables and project them into a unified space, which is then used to complete the task. This approach allows every interaction in the data to be effectively represented and utilized.

---

> ### Author Response · Authors · 2024-11-26
> **Response by authors**
>
> **Question 2. Questions with the experiments:**
>
> > Question 2a. Limited improvements on AV-MNIST and CMU-MOSEI (V+T).
>
> Thank you for pointing this out. The AV-MNIST and CMU-MOSEI (V+T) tasks are inherently challenging to learn. AV-MNIST is a synthetic dataset derived from MNIST, which has shown only modest improvements with previous methods compared to joint training. For example, prior work reports the best performance at 72.8%, while joint training (LF) achieves 71.7% [3]. In our experiments, our method surpasses joint training by 0.9%, validating the effectiveness of our approach.
> Furthermore, we replace this synthetic dataset with the real-world dataset, UCF-101, which includes Optical Flow (OF) and RGB modalities, as detailed in Table 2. Below, we present partial results, demonstrating the effectiveness of our method across real-world datasets:
>
> | Dataset  | Metric | RGB   | OF    | Joint  | Ensemble | DMI   |
> |----------|--------|-------|-------|--------|----------|-------|
> | UCF101   | ACC    | 76.9  | 67.8  | 78.8   | 82.3     | **84.2**  |
> |          | F1     | 76.1  | 67.6  | 78.0   | 81.8     | **83.9**  |
>
> Besides, sentiment analysis on the CMU-MOSEI dataset is also challenging. Previous studies have reported a significant improvement of 1.5% over joint learning in binary classification tasks across three modalities [3]. In this paper, we consider a tougher task with only two modalities. Our method achieves a 1.2% increase in the (A+T) setup for a three-way classification task, and demonstrates improvement in the (V+T) modality, whereas some comparison methods experience a drop in performance. This highlights both the complexity of learning in this setup and the effectiveness of our proposed approach.
>
> > Question 2b. Modifications specific to each modality in ResNet18.
>
> Thank you for your question. The modification involves adapting the network to accommodate modalities that differ from the typical three-channel RGB input. Specifically, for the audio modality, which has a single channel, and the optical flow modality, which consists of two channels, we modify the channel dimension of the first layer of ResNet18 to correspond with the channel dimensions of these modalities.
>
>
> > Question 2c. Details about MOSEI task.
>
> Thank you for the inquiry. The task conducted on MOSEI is a sentiment analysis task. These samples are divided into positive, negative, and neutral, following the setting in the previous study [4].
>
> > Question 2d. The performance of the unimodal method is missing.
>
> Thank you for your suggestion. We have supplemented the unimodal performance in Table 2 and Table 3 in the Experiment section, highlighting the improvements achieved by the multimodal methods.
>
> > Question 2e. Why DMI-TC and DMI-CD worse than DMI-FC.
>
> Thank you for your valuable comment. These three methods are based on our proposed DMI architecture. For DMI-FC, we adopt the principle of decomposition but replace the variational decomposition with a fully-connected layer. This can degrade the decouplement among modalities. DMI-TC and DMI-CD, on the other hand, represent partial decompositions within the DMI framework. The superior performance of DMI-FC showcases the significance of comprehensive decomposition. But with the decoupling among modalities, DMI can achieve better decomposition and thus achieve better performance.
>
> [1] P. P. Liang, Y. Cheng, X. Fan, C. K. Ling, S. Nie, R. Chen, Z. Deng, F. Mahmood, R. Salakhutdinov, and L.-P. Morency, “Quantifying & modeling multimodal interactions: An information decomposition framework,” in *Advances in Neural Information Processing Systems*, 2023.
>
> [2] A. A. Alemi, I. Fischer, J. V. Dillon, and K. Murphy, “Deep variational information bottleneck,” *arXiv preprint arXiv:1612.00410*, 2016.
>
> [3] P. P. Liang, Y. Lyu, X. Fan, Z. Wu, Y. Cheng, J. Wu, L. Chen, P. Wu, M. A. Lee, Y. Zhu et al., “Multibench: Multiscale benchmarks for multimodal representation learning,” *arXiv preprint arXiv:2107.07502*, 2021.
>
> [4] C. Hua, Q. Xu, S. Bao, Z. Yang, and Q. Huang, “Reconboost: Boosting can achieve modality reconcilement,” *arXiv preprint arXiv:2405.09321*, 2024.

---

> ### Author Response · Authors · 2024-11-29
> **Look forward to your feedback!**
>
> Dear Reviewer a4Ud,
>
> Thanks again for your insightful suggestions and comments. We would appreciate knowing if our responses have fully addressed your concerns. We are happy to answer any further questions or comments you may have.

---

> ### Author Response · Authors · 2024-12-02
> **Look forward to your feedback before deadline!**
>
> Dear Reviewer a4Ud,
>
> We would like to sincerely thank you for your time and effort in reviewing our paper and providing invaluable feedback. In response to your suggestions, we have clarified the method architecture and provided a more detailed explanation of the experimental setup, along with extended experimental results in the revised manuscript.
>
> If you have any further questions or require additional clarification, we would greatly appreciate it if you could inform us before the rebuttal period ends (**less than one day remaining**).
>
> Thank you once again for your insightful comments.
>
> Best regards,
>
> The Authors

---

> > ### Comment · Reviewer_a4Ud · 2024-12-02
> >
> > Dear Authors,
> >
> > Sorry for the late response and thanks for your rebuttal and hard work. Most of my concerns and questions regarding the method section are clarified. I appreciate it. However, I will maintain my current rating as I believe it still reflects my overall assessment.

---

> > > ### Author Response · Authors · 2024-12-03
> > > **Thank you!**
> > >
> > > We greatly appreciate your invaluable comments and positive feedback, and we are pleased to hear that the clarifications in the method section have addressed your questions.
> > >
> > > Best regards,
> > >
> > > The Authors

---

### Official Review · Reviewer_oH8U · 2024-11-04

**Soundness:** 2
**Presentation:** 3
**Contribution:** 2
**Rating:** 5
**Confidence:** 4

**Summary:**

The paper presents a method to explicitly decouple the redundancy (R), synergy (S), and uniqueness (U) of the information in a pair of modalities when doing multimodal learning. The proposed method first breaks down the information from each modality into task-specific (T) and task-irrelevant (V) information, followed by aligning the redundant parts of the information in T1 and T2 (R) while keeping the unique parts separate (U1 and U2), while trying to learn the synergy between V1 and V2. To improve the quality of the decomposition, the method first trains the modality encoders, following by freezing them and only training the decomposition module, and finally training the whole model end-to-end.

The paper presents results on audiovisual datasets like CREMA-D (for emotion recognition in an acted setting), AV-MNIST, Kinetics Sounds (for sound event detection), amd CMU MOSEI (for multimodal sentiment) and compares against baselines based on joint/ensemble learning, and those based on unimodal regulation and multimodal interaction.

**Strengths:**

I thank the authors for sharing their ideas.I think that some of the contributions of the paper are interesting, well thought out, and clearly presented:
- I found the clear breakdown and explanation of the R, U, and S type of multimodal interactions and how they can be explicitly optimized as part of a loss function while doing multimodal learning to be a strength of the paper. Intuitively, this idea makes sense, and has been illustrated well to the reader in the equations and the figures.
- The chosen baselines make sense.
- The presented results on the chosen datasets are strong when compared to the chosen baselines.
- The technical details of the experiments are clearly presented, and the setup is understandable.

**Weaknesses:**

While I think that the core motivation of the paper is solid, and the proposed approach makes sense, I believe where the paper in its current state falls short is the rigor and comprehensiveness of the experiments. The same method (along with the baselines), when demonstrated with more convincing technical choices/design would make for a much stronger paper.

- The paper presents itself as a general contribution to multimodal learning, however the demonstrated experiments are only on 3 limited audio-visual datasets (and CMU MOSEI in a limited way for audio-text and visual-text). I believe that only using these datasets is not as strong a result as, for example, also showing strong and comprehensive results on vision+text, audio+vision+text, or other sensory modalities like LIDAR, ultrasonic sensors, physiological sensors (EEG / eye tracking / EMG etc). The presented results are fine if the claim of the paper was slightly more focused to the audiovisual setting (or to only CMU MOSEI for multimodal sentiment analysis). However, I do not think the experiments are "Comprehensive" and "holistic" like the authors have claimed.
- Even within the audiovisual datasets, I have significant concerns about the methodology used to extract information from each modality. For example, in the Kinetics sound dataset (which itself is dominated by the audio modality), using 1 frame per second (and a total of 10 frames per video) is not ideal (as opposed to using every frame at 25 or 30 FPS).
For the CREMA-D datasets, the authors take 1 single frame from the entire video. This is incredibly limiting for affect recognition (especially on this specific dataset), due to not capturing the temporal dynamics of emotion (such as the onset-apex-offset (see https://www.researchgate.net/figure/Typical-development-of-a-facial-expression-with-onset-apex-and-offset-from-the-survey_fig1_326023674) dynamics). Using the temporal information from the video is critical here in my opinion (it just happens to be that in an acted dataset like CREMA-D the *acted* facial expression usually coincides with the center timestep of the video) to have a method that is more generally applicable.
- I also think that the scale of the chosen datasets was limited. For example, to demonstrate the approach on solely audiovisual interactions, there are alternative datasets like AVSpeech, LRS, LRW, Audioset, along with any of the datasets in https://openaccess.thecvf.com/content/CVPR2024/papers/Singh_Looking_Similar_Sounding_Different_Leveraging_Counterfactual_Cross-Modal_Pairs_for_Audiovisual_CVPR_2024_paper.pdf
- In my opinion, the choices of the backbones (ResNet18 and LeNet) are also sufficient for basic experiments, but not to draw holistic conclusions.

**Questions:**

- Are there any direct comparisons to the numbers presented in the baselines? E.g. Wu et al (2024) in MMML do an extensive set of results on CMU MOSI and CMU MOSEI, whereas their method’s baseline results in this paper are quite a bit lower. What specific differences exist in these two experimental settings?
- CMU MOSEI is by far the most appropriate dataset to demonstrate the method out of the chosen datasets, since it allows for multiple different modality pairs. Why did the authors choose to do the ablations on the smaller CREMA dataset and (larger) Kinetics sounds, which are both audiovisual only, instead of on MOSEI?
- Could the method be easily extended to interactions between additional modalities than just 2? For example, multiple visual streams (along with depth, LIDAR etc) in robotics, or just audio + vision + text in MOSEI etc. Why or why not?
- Certain typos (CRAME instead of CREMA) and broken links (e.g. hyperref link in Figure 1, citations in the tables) were broken

---

> ### Author Response · Authors · 2024-11-26
> **Response by authors**
>
> **Question 1. Lack for Holistic Experimental Results**
>
> Thank you for your valuable suggestions regarding the holistic validation of our methods. In response, we have incorporated extensive experiments in the revised manuscript, aligning with your recommendations. Below, we summarize the key questions addressed in these experiments:
>
> > Question 1a. More modalities.
>
> In addition to our initial audio-visual methods, we expand our experiments to include additional modalities. Specifically, we validate RGB + Optical Flow on the UCF101 dataset [1] for action recognition, Audio + Text on the UR-FUNNY dataset [2] for humor detection, and mRNA + methylation data on the ROSMAP dataset [3] for Alzheimer’s Disease diagnosis.
> We updated Table 2 to replace the synthetic dataset AV-MNIST with the UCF-101 dataset to better showcase the effectiveness of our method across real-world datasets, and add other experiments in Appendix B.3. These experimental results demonstrate the flexibility of our approach across different modalities and complex tasks.
>
> |  Dataset | UR-FUNNY |       |  UCF  |       | ROSMAP |       |
> |----------|:--------:|:-----:|:-----:|:-----:|:------:|:-----:|
> |  Metric  | ACC      | F1    | ACC   | F1    | ACC    | F1    |
> | Joint    | 63.8     | 63.7  | 78.8  | 78.0  | 84.0   | 83.8  |
> | Ensemble | 64.1     | 64.0  | 82.3  | 81.8  | 83.0   | 83.0  |
> | DMI      | **65.0**     | **64.7** | **84.2**  | **83.9**  | **84.9**   | **84.9**  |
>
> > Question 1b. Richer temporal information.
>
> This is an insightful question of considering richer temporal information. To address this, we expanded our application of the method across more frames within the CRMEA-D and KS datasets. Our findings indicate that increased richer temporal information enhances task performance to some degree. Moreover, our propose DMI paradigm still demonstrates improvements with this abdundant temporal information. Detailed results and analyses are provided in Appendix B.3 of the revised manuscript.
>
> | Temporal | CREMA-D-2frame |       | KS-8frame |       |
> |----------|----------------|-------|-----------|-------|
> | Metric   | ACC            | F1    | ACC       | F1    |
> | Joint    | 77.8           | 78.3  | 85.3      | 85.3  |
> | Ensemble | 77.7           | 78.2  | 87.1      | 87.1  |
> | DMI      | **78.5**           | **79.3**  | **87.5**      | **87.5**  |
>
> > Question 1c. Larger scale.
>
> After careful consideration of time and feasibility, we choose the VGGsound dataset[4], which encompasses over 210,000 entries across 309 categories. Detailed results are presented in the following table and further detailed in Appendix B. These results demonstrate that our DMI method significantly outperforms existing approaches on large-scale datasets.
>
>
> | VGGsound | Joint | Ensemble | DMI  |
> |----------|-------|----------|------|
> | ACC      | 55.1  | 56.7     | **58.5** |
> | F1       | 53.3  | 55.1     | **57.0** |
>
> > Question 1d. Different backbone.
>
> The chosen backbones, ResNet and LeNet, are widely utilized in multimodal research and are consistent with prior studies [5,6]. Also, we validate our method on the Hierarchical Multimodal Transformer backbone [7] as detailed in Table 3, which serves to further validate the scope of our approach.
> Additionally, we conducted expanded experiments on the KS dataset using ResNet34 and on the CMU-MOSEI dataset using an LSTM backbone. These experiments further verify the effectiveness of our method. Detailed results are presented in Appendix B.3.
>
> | Dataset  | MOSEI |      |    KS    |      |
> |----------|:-----:|:----:|:--------:|:----:|
> | Backbone |  LSTM |      | ResNet34 |      |
> | Metric   | ACC   | F1   | ACC      | F1   |
> | joint    | 62.4  | 62.2 | 86.0     | 85.8 |
> | ensemble | 62.0    | 61.7 | 86.8     | 86.3 |
> | DMI      | **62.9**  | **62.9** | **87.8**     | **87.7** |
>
> **Question 2. Specific differences in MMML.**
>
> Thank you for your comment. To ensure a fair comparison across different methods, we standardized the backbone across all approaches, followed by [6], applying distinct strategies of different methods to this common framework. For the MMML approach, we incorporated its fusion module on the aligned backbone. This module includes both an attention mechanism and a multi-loss strategy.

---

> ### Author Response · Authors · 2024-11-26
>
> **Question 3. Ablation study setting.**
>
> Thank you for your comment. The primary goal of our ablation study is to verify the effectiveness of each module within our proposed decomposition method. Specifically, we aim to understand how these modules perform across various datasets and various data scales.
> We conducted extensive experiments on the CMU-MOSEI dataset, which includes both audio and text (A+T) and Visual+Text (V+T) modalities. The results in the Table below also demonstrate the effectiveness of each module. We have updated the results in Appendix B.3.
>
> | Dataset  | MOSEI(A+T) | MOSEI(V+T) |
> |----------|------------|------------|
> | DMI-FC   | 62.6       | 61.6       |
> | DMI-CD   | 61.4       | 63.2       |
> | DMI-TD   | 61.3       | 62.2       |
> | DMI      | **63.1**       | **63.4**      |
>
> **Question 4. Extended to modality larger than 2.**
>
> Thank you for this valuable suggestion. Our proposed Decomposition-based Multimodal Interaction learning (DMI) approach is adaptable to scenarios involving three modalities. By implementing only the Task-related Decomposition on DMI (DMI-TD, illustrated in Figure 4), we can extend our framework to accommodate three modalities.
> We have conducted empirical evaluations on two datasets that each incorporate three modalities: MOSEI, which includes Visual (V), Audio (A), and Text (T) modalities, and UCF101, which consists of RGB, Optical Flow (OF), and Frame Difference (Diff) modalities. Detailed results of these experiments are presented in Appendix B.3.3, verifying that our method remains effective when extending to three modalities.
>
>
> |  Dataset | MOSEI (V+A+T) |       | UCF (RGB+OF+Diff) |       |
> |:--------:|:-------------:|:-----:|:-----------------:|:-----:|
> |  Metric  |      ACC      |   F1  |        ACC        |   F1  |
> |   joint  |     63.3      | 63.2  |       78.6        | 78.2  |
> | ensemble |     63.4      | 62.7  |       84.4        | 83.9  |
> |  DMI-TD  |     **64.3**      | **64.5**  |       **84.8**        | **84.2**  |
>
>
> [1] K. Soomro, “Ucf101: A dataset of 101 human actions classes from videos in the wild,” *arXiv preprint arXiv:1212.0402*, 2012.
>
> [2] M. K. Hasan, W. Rahman, A. Zadeh, J. Zhong, M. I. Tanveer, L.-P. Morency et al., “Ur-funny: A multimodal language dataset for understanding humor,” *arXiv preprint arXiv:1904.06618*, 2019.
>
> [3] P. L. De Jager, Y. Ma, C. McCabe, J. Xu, B. N. Vardarajan, D. Felsky, H.-U. Klein, C. C. White, M. A. Peters, B. Lodgson et al., “A multi-omic atlas of the human frontal cortex for aging and alzheimer’s disease research,” *Scientific data*, vol. 5, no. 1, pp. 1–13, 2018.
>
> [4] H. Chen, W. Xie, A. Vedaldi, and A. Zisserman, “Vggsound: A large-scale audio-visual dataset,” in *ICASSP 2020-2020 IEEE International Conference on Acoustics, Speech and Signal Processing* (ICASSP). IEEE, 2020, pp. 721–725.
>
> [5] Y. Fan, W. Xu, H. Wang, J. Wang, and S. Guo, “Pmr: Prototypical modal rebalance for multimodal learning,” in *Proceedings of the IEEE/CVF Conference on Computer Vision and Pattern Recognition*, 2023, pp. 20 029–20 038.
>
> [6] P. P. Liang, Y. Lyu, X. Fan, Z. Wu, Y. Cheng, J. Wu, L. Chen, P. Wu, M. A. Lee, Y. Zhu et al., “Multibench: Multiscale benchmarks for multimodal representation learning,” *arXiv preprint arXiv:2107.07502*, 2021.
>
> [7] P. Xu, X. Zhu, and D. A. Clifton, “Multimodal learning with transformers: A survey,” *IEEE Transactions on Pattern Analysis and Machine Intelligence*, 2023.

---

> > ### Comment · Reviewer_oH8U · 2024-11-26
> > **Insufficient rigor remains**
> >
> > I thank the authors for their response and the updated version of the paper.
> >
> > Positives in the rebuttal:
> > - Inclusion of other modalities
> > - Slightly better performance compared to baselines on new results
> >
> > Unfortunately, most of my concerns persist:
> > - Doing eval on CREMA with only 2 frames, while better than one frame, still has the same issue that I originally described. This does not properly represent the affective information in a general video, and could be seen as 'cheating' in an acted dataset like CREMA where the facial expression will contain the prototypical emotional expression at a specific time in the video.
> >
> > - The apple-to-apple comparison with the protocol established in the baseline still remains.
> >
> > > Question 2. Specific differences in MMML.
> > > Thank you for your comment. To ensure a fair comparison across different methods, we standardized the backbone across all approaches, followed by [6], applying distinct strategies of different methods to this common framework. For the MMML approach, we incorporated its fusion module on the aligned backbone. This module includes both an attention mechanism and a multi-loss strategy.
> >
> > Why not use the same settings from the baseline instead?
> >
> > - The backbones are still insufficient
> >
> > > The chosen backbones, ResNet and LeNet, are widely utilized in multimodal research
> >
> > These are fine for basic experiments, but there are significantly stronger choices like ViT's. Also, the results in Table 3 are much lower than the ones reported in the original papers which suggests a lot of room for improvement in the experimental protocol.

---

> > > ### Author Response · Authors · 2024-11-27
> > > **Response by authors**
> > >
> > > Thank you for your thoughtful responses! Below are our replies to your comments:
> > >
> > > > Prototypical emotional expression in CREMA-D dataset.
> > >
> > > Thank you for your valuable suggestion. We have conducted experiments on the CREMA-D dataset using 8 frames and observed that increasing the number of frames significantly enhances performance on this emotion recognition task. The experimental results validate the effectiveness of our method when incorporating varying amounts of temporal information. These findings are discussed in Appendix B.3.4 of the revised manuscript.
> > >
> > >
> > > | Temporal | CREMA-D-1Frames |  | CREMA-D-2Frames |  | CREMA-D-8Frames |  |
> > > |:---:|:---:|:---:|:---:|:---:|:---:|:---:|
> > > | Metric | ACC | F1 | ACC | F1 | ACC | F1 |
> > > | Joint | 70.2  | 71.0  | 77.8  | 78.3  | 85.5  | 85.9  |
> > > | Ensemble | 68.8  | 69.5  | 77.7  | 78.2  | 86.6  | 87.0  |
> > > | DMI | **73.1** | **73.8** | **78.5** | **79.3** | **87.5** | **87.9** |
> > >
> > > > The settings of our experiment, and results in Table 3.
> > >
> > > We appreciate your insightful comment regarding the experimental settings. Our method is compared against a variety of approaches, including methods with unimodal regulation (OGM, PMR, AGM) and those with architecture designs for fusion in interaction captioning (MBT, MIB, QMF, MMML). Conducting a fair comparison is challenging due to the differences in datasets and settings across these methods. To ensure a fair comparison, we follow the MultiBench benchmark [1], which is widely used. Following [1], we utilize a Transformer backbone on pre-extracted features, standardizing the backbones to enable consistent and fair comparison across different fusion methods. Additionally, we have clarified the criteria for backbone selection in Section 4.1 of the revised manuscript.
> > >
> > > > ViT backbones.
> > >
> > > Thank you for raising this point. We have considered the incorporation of the Vision Transformer (ViT) backbone in our experiments. Specifically, both the visual and audio modalities are processed using a 4-layer ViT structure on the Kinetic-Sound dataset. The results for this experiment are presented in Table 3, as shown below. We clarity the backbone in Section 4.1 of the revised manuscript.
> > >
> > > | Dataset | Metric | Audio | Visual  | Joint | Ensemble | OGM | PMR  | AGM  | MBT | MIB | QMF | MMML | DMI |
> > > |:---:|:---:|:---:|:---:|:---:|:---:|:---:|:---:|:---:|:---:|:---:|:---:|:---:|:---:|
> > > | Kinetic-Sound | ACC | 50.5  | 50.9  | 67.9  | 69.3  | 68.9  | 68.0  | 68.9  | 69.9  | 63.1  | 70.6  | 65.3  | **70.8**  |
> > > |  | F1 | 50.3  | 50.5   | 67.6  | 69.2  | 68.6  | 68.2  | 69.1  | 69.9  | 62.9  | 70.3  | 65.5  | **71.4**  |
> > >
> > > [1] P. P. Liang, Y. Lyu, X. Fan, Z. Wu, Y. Cheng, J. Wu, L. Chen, P. Wu, M. A. Lee, Y. Zhu et al., “Multibench: Multiscale benchmarks for multimodal representation learning,” *arXiv preprint arXiv:2107.07502*, 2021.

---

> ### Author Response · Authors · 2024-11-29
> **Look forward to your feedback!**
>
> Dear oH8U,
>
> Thank you for your thoughtful review and constructive feedback. We have carefully reviewed your comments and made the necessary adjustments. We would appreciate it if you could confirm whether the revisions address your concerns. We would be happy to provide additional information if any further clarification is required.

---

> ### Author Response · Authors · 2024-12-02
> **Request for Feedback Before Rebuttal Deadline**
>
> Dear Reviewer oH8U,
>
> We would like to sincerely thank you for reviewing our paper and providing valuable feedback. In response to your suggestion, we have added experimental results and clarified the experimental setting in the revised manuscript.
>
> Please feel free to reach out if you have any further questions or require additional clarification before the rebuttal period concludes (**less than one day remains**).
>
> Thank you once again for your insightful comments.
>
> Best regards,
>
> The Authors

---

### Official Review · Reviewer_HrvF · 2024-11-10

**Soundness:** 2
**Presentation:** 2
**Contribution:** 3
**Rating:** 5
**Confidence:** 3

**Summary:**

This paper investigates multimodal interaction through the lens of decomposition. The authors argue that the existing learning paradigms, such as joint learning and modality ensemble, struggle to handle all types of interaction effectively, leading to generalization issues. To address this, the authors propose a new paradigm called decomposition-based multimodal interaction (DMI) learning. DMI decomposes multimodal interaction into separate interaction types and applies a new training strategy to enhance learning across these interactions.

**Strengths:**

* The authors tackle the important problem of multimodal learning through mutual information decomposition.
* Improvements over state-of-the-art.
* The design of the Interaction Decomposition Module is very creative and aims to completely break down mutual information through learning. Figure 3, which shows the Interaction Decomposition Module, is very informative.

**Weaknesses:**

* I'm confused by the results in Table 1 and the associated section 3.3. Reading through the proof for Lemma 3.3 from Appendix A.3, it looks like the N = 2n relationship comes from how the authors constructed the new dataset by separating X(1) and X(2): S(1) = {X(1), Φ(2), Y}, S(2) = {Φ(1), X(2), Y}. However, these are the **same** data, just arranged differently to emulate the unimodal MI instead of multimodal MI to create the bounds in Eqs (25) and (26); as a side effect, there are now twice as many samples. But the underlying data stays exactly the same -- we don't need more data to train each model paradigm. What does this result mean in practice?
* The model architecture seems unclear. There isn't much information available beyond Figure 3, and it would be difficult to understand or reproduce this work with the information given.

**Questions:**

* Table 1 uses this form of big-O notation $O(1/\sqrt{N})|_{N=2n}$. Does $N=2n$ simply mean "set N to 2n" (I've not seen this condition before)? Since big-O notation is asymptotic, what difference does it make between $N=n$ vs. $N=2n$?
* My takeaway from this paper is "interaction decomposition improves multimodal interaction learning". Is this what the paper is trying to convey?
* Are joint learning and ensemble learning the only paradigms available? Are there other paradigms that address the proposed problem?

---

> ### Author Response · Authors · 2024-11-26
> **Response by authors**
>
> **Question 1. Expression in formulation**
>
> > Question 1a. Theoretical contributions of Lemma 3.3.
>
> Thank you for your insightful comment. Lemma 3.3 describes that under redundancy-type interactions, the modality ensemble approach can reduce the generalization gap between the theoretical mutual information, $I(Z;Y|\textbf{c})$, and the empirical mutual information, $I_S(Z;Y|\textbf{c})$. The ensemble paradigm aims to solely learn from each unimodality to complete the task. Consequently, we introduce $\tilde{S}$ (referenced in Lemma 3.3) to provide a more accurate representation of the learning paradigm. Due to the nature of redundancy interaction, both modality contains sufficient task-related information. Therefore, every sample within $\tilde{S}$ contributes to training the multimodal task.  With $2n$ samples available in $\tilde{S}$, according to Proposition 3.2, a reduction in the generalization gap is achieved. This reduction in the generalization gap implies that enhancing the empirical mutual information contributes to an increase in the theoretical mutual information, similar to the Probably Approximately Correct (PAC) learning theory. Hence, we present in Table 1 that modality ensemble achieves a tighter upper bound than joint learning under redundancy-type interactions.
>
> > Question 1b. Confusion of big-O notion
>
> Thank you for your invaluable comment. Our analysis of Big-O notation is to illustrate the differences in the upper bounds of the generalization gap (as defined in Equation 8) between the modality ensemble and joint learning paradigms. This presentation may have lacked clarity. To address this, we have revised the relevant equation to improve both its clarity and simplicity. The revised of Table 1 is presented below:
> |        	|              Redundancy              	|           Uniqueness          	|                               Synergy                      	|
> | ---------- | ------------- | ---------------------- | -------------- |
> |   Joint  	| $\leq \xi + \sqrt{\frac{\omega}{n}} $ 	| $\leq \xi + \sqrt{\frac{\omega}{n}} $ 	|                     $\leq \xi + \sqrt{\frac{\omega}{n}} $                     	|
> | Ensemble 	|   $\leq \xi + \sqrt{\frac{\omega}{2n}} $   	| $\leq \xi + \sqrt{\frac{\omega}{n}} $ 	| $\geq \max \left(I^{syn}_S(Z^{(1)}; Y), I^{syn}_S(Z^{(2)}; Y)\right)$ 	|
>
> The revised table now more accurately represents the differences, emphasizing that the main variance in redundancy is the factor $\sqrt{\frac{\omega}{n}}$ for joint learning compared to $\sqrt{\frac{\omega}{2n}}$ for modality ensemble. Further details of the modifications are discussed on Page 5, Section 3.3 of our manuscript. Thank you again for this invaluable comment.
>
> **Question 2. Explanation of model architecture.**
>
> Thank you for raising this point. The architecture of our proposed DMI (see Figure 3) consists of unimodal encoder $\phi^{(m)}$ to obtain unimodal representation $Z^{(m)}$, and two decomposition modules, decomposing the representation into different interactions $R, U^{(1)}, U^{(2)}, S$. The unimodal encoder varies for different tasks. The decomposition module is architecture like Variational Autoencoder (VAE). Each decomposition module is structured on a VAE framework, where the encoders, composed of Multi-Layer Perceptrons (MLPs), predict the mean and variance. Conversely, the decoders are designed as multilayer networks to ensure minimal information loss during the decomposition process. The alignment of features across modalities is enforced by minimizing the Kullback-Leibler (KL) divergence between the corresponding distributions. We add more detailed descriptions in Appendix B.2.
>
> **Question 3. The idea paper conveys.**
>
> Thank you for your insightful observation. In addition to your mention of how **interaction decomposition improves multimodal interaction learning**, we have provided a comprehensive theoretical analysis that underscores **the importance of considering various multimodal interactions within multimodal learning**. This analysis explains the crucial role that learning from holistic interactions helps better multimodal learning performance. It also illuminates the underlying mechanics of our interaction decomposition method, further justifying its application.

---

> ### Author Response · Authors · 2024-11-26
> **Response by authors**
>
> **Question 4. Other paradigms over joint learning and modality ensemble.**
>
> Thank you for highlighting these points. In supervised learning that involves multiple modalities, the central challenge lies in leveraging multimodal data effectively to accomplish multimodal tasks. According to previous literature [1], joint learning of multiple modalities in a shared space and separate learning within each modality's own space are two primary paradigms in multimodal learning. Based on this, two prevalent paradigms—joint learning and modality ensemble—represent different approaches to this challenge. The modality ensemble approach utilizes individual modalities independently to complete tasks, whereas joint learning integrates all modalities collectively for task completion. Additionally, in the domain of self-supervised learning, alignment, and contrastive learning are commonly employed paradigms. The insights gained from analyses within these paradigms are invaluable and will be considered in our future research.
>
> [1] T. Baltrusaitis, C. Ahuja, and L.-P. Morency, “Multimodal machine learning: A survey and taxonomy,” *IEEE Transactions on pattern analysis and machine intelligence*, vol. 41, no. 2, pp. 423–443, 2018.

---

> ### Author Response · Authors · 2024-11-29
> **Look forward to your feedback!**
>
> Dear Reviewer HrvF,
>
> Thank you once again for your valuable comments and suggestions. We would appreciate your confirmation on whether our responses have addressed your concerns. Please let us know if you have any further questions or comments.

---

> ### Author Response · Authors · 2024-12-02
> **Request for Feedback Before Rebuttal Deadline**
>
> Dear Reviewer HrvF,
>
> We would like to sincerely thank you for reviewing our paper and providing invaluable feedback. Your insights have greatly contributed to enhancing the quality of the manuscript.
>
> If there are any further points that require clarification, we would be grateful if you could let us know before the end of the rebuttal period (**less than 1 day remaining**).
>
> Thank you once again for your thoughtful and constructive comments.
>
> Best regards,
>
> The Authors

---

### Author Response · Authors · 2024-11-26
**Thanks for reviewing our work**

Dear reviewers, we sincerely appreciate all your constructive comments and encouraging remarks (e.g., Reviewer HrvF: *Interaction Decomposition Module is very creative...informative*; Reviewer oH8U: *clear breakdown and explanation of the R, U, and S*; Reviewer a4Ud: *promising performance*; Reviewer 9UkZ: *Novel theoretical analysis and New learning paradigm for interaction learning*). Although some absence of experimental details and the need for more comprehensive experiments raised concerns among the reviewers, we have carefully addressed these points during the revision process, making substantial enhancements to strengthen these aspects. Below, we summarize the key contributions and revisions made to our manuscript:

**Key Contributions:**

In this paper, we have provided a comprehensive theoretical analysis that emphasizes **the importance of considering various multimodal interactions in multimodal learning**. This analysis highlights the crucial role that learning from holistic interactions plays in improving multimodal learning performance. Building upon this, we introduce a novel learning paradigm, the Decomposition-based Multimodal Interaction learning(DMI) framework, **which leverages interaction decomposition to enhance multimodal learning**. Our method decomposes multimodal interactions into three distinct types: redundancy, uniqueness, and synergy, enabling the model to effectively learn from each type of interaction.

**Revisions:**

*Explicit Elaboration about Method and Experimental Details*:

We have refined unclear explanations and notations in our analysis (Section 3.3, Table 1), addressed inaccuracies in our model descriptions (Section 3.4 and Appendix B.2), and provided a detailed description of the synthetic dataset used (Appendix B.4).

*Expanded Experimentation*:

We have conducted extensive experiments involving larger datasets (VGGsound), multiple modalities (UCF RGB + Optical Flow, ROSMAP mRNA + METH), and different tasks (UR-FUNNY for humor detection). We have expanded our studies to include more than two modalities (UCF & MOSEI), introduced richer temporal information, and changed the backbone validated on datasets like KS. Detailed descriptions of these experiments are provided in Appendix B.3.

---

### Meta-Review · Area_Chair_DtAx · 2024-12-20

**Metareview:**

This paper studies how to learn representations capturing different multimodal interactions during multimodal learning. They propose information-theoretical evidence that learning all types of interactions (redundancy, uniqueness, synergy) is necessary for good performance and shows that naive joint and ensemble learning cannot learn all types of interactions equally well. Motivated by this finding, they proposed a method called Decomposition-based Multimodal Interaction learning (DMI) to decompose multimodal information into different types of interactions learned via a three-phase training.

The authors generally appreciated the theoretical framework, the proposed method, and promising performance on several datasets.

There were concerns initially regarding lack of multimodal datasets in the experimental, lack of clarity in the model architecture and overall in the proposed method, and requests for further validation experiments. The authors provided experiments on more datasets and added several useful analyses in their rebuttal, which 2 reviewers appreciated, and gave a score of marginal accept. 2 reviewers stayed with their scores of marginal reject. Due to the exactly borderline split of reviews, I read the paper in detail, all reviews, and all discussions. I am inclined to lean towards rejection, since there is a severe issue: the paper offers almost no details as to the model architecture, training objectives, and overall algorithm (just as reviewer HrvF pointed out). There is a lot of math to show that the method can work in theory to capture different information-theoretic quantities, but how these are learned in practice is completely vague. The authors also respond in extremely vague terms, saying 'the unimodal encoder varies for different tasks' and 'the decomposition module is architecture like Variational Autoencoder (VAE)'. This is an unacceptable level of rigor and detail for a conference like ICLR, and I would recommend the authors to be extremely upfront about all the design decision, modeling architectures, and training objectives used in the work.

**Additional Comments On Reviewer Discussion:**

During the discussion process, reviewers a4Ud and 9UkZ confirmed that their concerns were addressed and gave a final score of 6.

Reviewer oH8U and the authors also engaged in several back-and-forth discussions, primarily about results on new datasets, different backbone architectures, and experimental settings to ensure fair comparison. Reviewer oH8U maintained their score of 5, and from what I've seen, the authors provided many more results during the discussion, but most of these results are only 1-2% better than the baselines and ablations, so without rigorous statistical tests I'm not sure if these results are significant.

For reviewer HrvF, they also maintained their score of 5, and after looking through the rebuttals I find that the concern they raised on unclear model architecture and overall poor clarity in the paper remains.

---

### Decision · Program_Chairs · 2025-01-22

Reject